# Ezrin enrichment on curved membranes requires a specific conformation or interaction with a curvature-sensitive partner

Feng-Ching Tsai[1,2†]*, Aurelie Bertin[1,2†], Hugo Bousquet[2,3], John Manzi[1,2], Yosuke Senju[4], Meng-Chen Tsai[5,6], Laura Picas[7], Stephanie Miserey-Lenkei[2,3], Pekka Lappalainen[4], Emmanuel Lemichez[6], Evelyne Coudrier[2,3‡]*, Patricia Bassereau[1,2‡]*

[1]Laboratoire Physico Chimie Curie, Institut Curie, PSL Research University, CNRS UMR168, Paris, France; [2]Sorbonne Université, Paris, France; [3]Compartimentation et dynamique cellulaire, Institut Curie, PSL Research University, CNRS UMR144, Paris, France; [4]Program in Cell and Molecular Biology, Institute of Biotechnology, University of Helsinki, Helsinki, Finland; [5]Université Côte d'Azur, CNRS, Institut de Pharmacologie Moléculaire et Cellulaire, Valbonne, France; [6]Département de Microbiologie, Unité des Toxines Bactériennes, Université Paris Descartes, Institut Pasteur, Paris, France; [7]Institut de Recherche en Infectiologie de Montpellier (IRIM), CNRS UMR 9004, Montpellier, France

*For correspondence:
feng-ching.tsai@curie.fr (FT);
Evelyne.Coudrier@curie.fr (EC);
patricia.bassereau@curie.fr (PB)

[†]These authors contributed equally to this work
[‡]These authors also contributed equally to this work

**Abstract** One challenge in cell biology is to decipher the biophysical mechanisms governing protein enrichment on curved membranes and the resulting membrane deformation. The ERM protein ezrin is abundant and associated with cellular membranes that are flat, positively or negatively curved. Using *in vitro* and cell biology approaches, we assess mechanisms of ezrin's enrichment on curved membranes. We evidence that wild-type ezrin (ezrinWT) and its phosphomimetic mutant T567D (ezrinTD) do not deform membranes but self-assemble anti-parallelly, zipping adjacent membranes. EzrinTD's specific conformation reduces intermolecular interactions, allows binding to actin filaments, which reduces membrane tethering, and promotes ezrin binding to positively-curved membranes. While neither ezrinTD nor ezrinWT senses negative curvature alone, we demonstrate that interacting with curvature-sensing I-BAR-domain proteins facilitates ezrin enrichment in negatively-curved membrane protrusions. Overall, our work demonstrates that ezrin can tether membranes, or be targeted to curved membranes, depending on conformations and interactions with actin and curvature-sensing binding partners.
DOI: https://doi.org/10.7554/eLife.37262.001

## Introduction

Attachment of the plasma membrane or the membrane of organelles to the actin cytoskeleton network is an important feature that controls cell shape and function. Ezrin, a member of the ezrin-radixin-moesin (ERM) protein family, plays a crucial role in linking the actin cytoskeleton to membranes. Ezrin is involved in various physiological processes including cell migration, cell signaling and the establishment of cell polarity (*McClatchey and Fehon, 2009*) (*Fehon et al., 2010*) (*Arpin et al., 2011*). Ezrin consists of an N-terminal FERM (band 4.1, ezrin, radixin, moesin) domain that binds to phosphatidylinositol 4,5-bisphosphate (PIP$_2$) lipids and membrane-associated binding partners, a structurally uncharacterized α-helical domain, and a C-terminal ERM association domain (C-ERMAD)

that interacts with the FERM domain and the actin cytoskeleton. The head-to-tail intramolecular interaction between the FERM domain and the C-ERMAD keeps ezrin in a closed configuration where the actin binding site is masked (*Bretscher et al., 1995*). "Opening up" of ezrin requires the FERM domain to bind to PIP$_2$, followed by the phosphorylation of the C-ERMAD (*Fievet et al., 2004*) (*Pelaseyed et al., 2017*). So far, the conformations of non-phosphorylated ezrin and of its phosphomimetic mutant have been studied either in solution, with or without PIP$_2$ micelles (*Pelaseyed et al., 2017*) (*Jayasundar et al., 2012*), or on bare solid substrates (*Liu et al., 2007*). A recent paper reports a conformational change of the pseudophosphorylated mutant of ezrin upon binding to PIP$_2$-containing supported bilayers (*Shabardina et al., 2016*), but a detailed analysis at high resolution at the nanometer scale of the different membrane-bound conformations of ezrin is still missing.

In cells, ezrin is one of the most abundant proteins at the plasma membrane (*Neisch and Fehon, 2011*). Given the actin-membrane linking function of ezrin and the mechanical action of actin on membranes, it is essential for cells to precisely regulate the membrane localization of ezrin. Ezrin is enriched in actin-rich plasma membrane structures such as microvilli (*Sauvanet et al., 2015*), filopodia (*Osawa et al., 2009*) and at the edge of bacterial toxin-induced transendothelial cell tunnels (*Stefani et al., 2017*). In mutant mice lacking ezrin, enterocytes show thicker, shorter and misoriented microvilli, suggesting that ezrin contributes to microvilli morphology (*Casaletto et al., 2011*). In these plasma membrane protrusions ezrin is located at the cytosolic side wherein the membrane has a negative mean curvature. Therefore, this suggests ezrin has a strong affinity for negatively curved membranes; in other words, ezrin may be a negative membrane curvature-sensing protein. However, ezrin is also associated with some intracellular vesicles including endosomes (*Chirivino et al., 2011*), wherein the membranes have a positive mean membrane curvature. Moreover, ezrin is found at flat regions of the plasma membrane, such as at the cortex-membrane interface and at the surface of membrane blebs (*Charras et al., 2006*). How the same protein can be a positive and a negative membrane curvature sensor is a conundrum that remains to be deciphered.

The complexity of studying ezrin-membrane interaction *in cellulo*, due to the presence of actin and cellular organelles, can be circumvented by using purified proteins and model membranes with controlled membrane curvature. In this study we combine cryo-electron microscopy (cryo-EM) and mechanical measurements using model membranes with cell biology approaches to compare wild type ezrin (ezrinWT) and its phosphomimetic mutant T567D (ezrinTD) for correlating ezrin conformation with its association to curved membranes. We show that: (1) both ezrinWT and ezrinTD assemble in an anti-parallel manner to tether adjacent membranes, but tethering is modulated by filamentous actin (F-actin); (2) only the phosphomimetic mutant senses positive membrane curvature, likely due to its different conformation compared to ezrinWT; and (3) neither ezrinWT nor ezrinTD senses negative membrane curvature. Although *in vivo* most of ezrin is associated with membrane protrusions having negative membrane curvature, we show that the enrichment of ezrin and its phosphomimetic mutant on negatively curved membranes is facilitated by their direct interaction with curvature-sensing proteins, for example inverse-Bin-Amphiphysin-Rvs (I-BAR) domain proteins. Altogether our data demonstrates the mechanisms for enriching ezrin on curved membranes, and reinforces the view of ezrin as a membrane-cytoskeleton linker and a scaffolding protein rather than a membrane shaper.

## Results

### Conformations of ezrin bound to PIP$_2$-containing membranes revealed at the nanometer scale

To assess how the phosphorylation influences ezrin conformation and its binding to PIP$_2$ membranes, we purified recombinant wild type ezrin with a histidine (His) tag (His-ezrinWT) and a phosphomimetic mutant where the threonine at position 567 was replaced by an aspartate (His-ezrinTD), mimicking the open configuration of ezrin (*Figure 1—figure supplement 1A*) (*Fievet et al., 2004*). After proteolysis of the His tag, ezrinWT and ezrinTD were labeled with Alexa dyes for detection by confocal fluorescence microscopy (*Figure 1—figure supplement 1B*). To measure ezrinWT or ezrinTD binding to membranes independently of membrane curvature, we used giant unilamellar vesicles (GUVs) having diameters of around 5 µm or more (thus flat at the scale of ezrin molecules) consisting

of brain total lipid extract with or without 5 mole % PIP$_2$. We measured the fluorescence signals of the labeled ezrin on GUVs by two independent techniques, confocal microscopy and flow cytometry. In agreement with previous reports, we found that ezrinTD and ezrinWT do not bind to GUVs lacking PIP$_2$ (*Figure 1—figure supplement 1C–F*) (*Carvalho et al., 2008*). When bound to PIP$_2$-containing GUVs, homogeneous ezrin fluorescence signals were observed on the membranes for both ezrinTD and ezrinWT (*Figure 1—figure supplement 1G* top). Ezrin-decorated GUVs were globally spherical without optically detectable membrane deformation at bulk ezrin concentrations ranging from 20 nM to 4 µM. To compare the binding affinities of ezrinTD and ezrinWT, we measured the absolute membrane surface fraction of fluorescent ezrin on GUV membranes, $\Phi_v$, at various bulk ezrin concentrations, $C_{bulk}$ (*Figure 1—figure supplement 1G* bottom) (*Sorre et al., 2012*). By assuming a non-cooperative binding reaction, we fitted the binding curves with a hyperbola, $\Phi_v = \Phi_{max} \times C_{bulk}/(C_{bulk} + K_d)$ , where $\Phi_{max}$ is the maximum membrane surface fraction of ezrin and $K_d$ is the dissociation constant (*Pollard, 2010*). For $\Phi_{max}$ = 12%, $K_d$ is equal to 1.2 µM and 4.2 µM for ezrinTD and ezrinWT, respectively. The estimated $K_d$ is comparable to the previously reported value obtained using similar free-standing membranes, large unilamellar vesicles (LUVs) (*Blin et al., 2008*), but is higher than reported values using solid-supported lipid bilayers (SLBs) (*Bosk et al., 2011*). Nonetheless, ezrinTD has a higher binding affinity for PIP$_2$ than ezrinWT, as previously reported (*Fritzsche et al., 2014*) (*Zhu et al., 2007*), confirming the phosphomimetic substitution of T567 by an aspartic acid facilitates the binding of ezrin to PIP$_2$.

Having checked how PIP$_2$ influences ezrin binding, we investigated how ezrin organizes on membranes at the nanometric level and in an aqueous ionic environment by using LUVs combined with cryo-EM. We prepared LUVs with diameters ranging between 100 and 500 nm by using the detergent removal method (*Rigaud et al., 1998*). In the absence of ezrin, LUVs were spherical and unilamellar (*Figure 1A*). In the presence of ezrinTD or ezrinWT, to our surprise, we observed regular stacks of membranes tethered together by ezrinTD or ezrinWT (*Figure 1B*). Three-dimensional reconstructions from cryo-tomography revealed that the lipid stacks are plate-like (*Video 1* and *Video 2*). To enhance the signal-to-noise ratio, we performed two-dimensional (2D) single particle analysis by selecting pieces of stacks comprising two bilayers and the protein material between them. The class-averages revealed densely packed globular domains on the lipid layers that are 4.5 nm apart from the membrane (as measured from the center of the globular domains to the closest lipid leaflet) (*Figure 1C*). Further 2D image analysis centered on the globular domains revealed their dimensions to be 5.4 ± 0.6 nm long by 3.5 ± 0.5 nm wide for ezrinTD and 5.3 ± 0.5 nm long by 2.9 ± 0.3 nm wide for ezrinWT (mean ± standard deviation, averaged from 21 and 7 classes for ezrinTD and ezrinWT, respectively) (*Figure 1D*). This is in good agreement with the reported FERM domain structure of ezrin (*Smith et al., 2003*), suggesting that these globular domains correspond to the FERM domain. The α-helical domain and C-ERMAD were less resolved under these experimental conditions. Nevertheless, we found that the distances measured between the centers of the globular domains of the two opposing molecules sandwiched between the lipid layers were different: 24.1 ± 1.3 nm for ezrinTD and 28.7 ± 1.2 nm for ezrinWT (mean ± standard deviation, averaged from 10 classes for both ezrinTD and ezrinWT, respectively) (*Figure 1E*). These distances are comparable with the proposed model of the fully open ERM proteins (24–32 nm) (*Fehon et al., 2010*) (*Jayasundar et al., 2012*) (*Phang et al., 2016*), and with lengths measured for ezrin (15–45 nm) (*Liu et al., 2007*) in its phosphorylated/open states and in the absence of PIP$_2$. Thus, our observations suggest that on the tethered membranes, both ezrinTD and ezrinWT have a different but open conformation. By measuring the frequency of membrane tethering at the same bulk ezrin concentration, we observed that ezrinTD induces more membrane tethering as compared to ezrinWT (*Figure 1—figure supplement 2A*, p<0.0001 by z-test). We further determined the minimal bulk ezrin concentration for inducing membrane tethering, ~30 nM and ~60 nM, for ezrinTD and ezrinWT, respectively (*Figure 1—figure supplement 2B*). Taken together, our results demonstrate that on PIP$_2$-containing membranes, both ezrinWT and ezrinTD self-assemble into brush-like pairs that dramatically disrupt and reorganize LUVs into bilayer stacks. Moreover, ezrinTD and ezrinWT molecules exhibit different lengths that reflect their distinct conformations.

If ezrin and its phosphomimetic mutant have different conformations on membranes, the binding energy between two tethered membranes should be different. To test this hypothesis, we performed experiments using a dual micropipette aspiration assay. Two GUVs decorated with ezrin were brought in contact while controlling the membrane tension of both GUVs by the micropipette

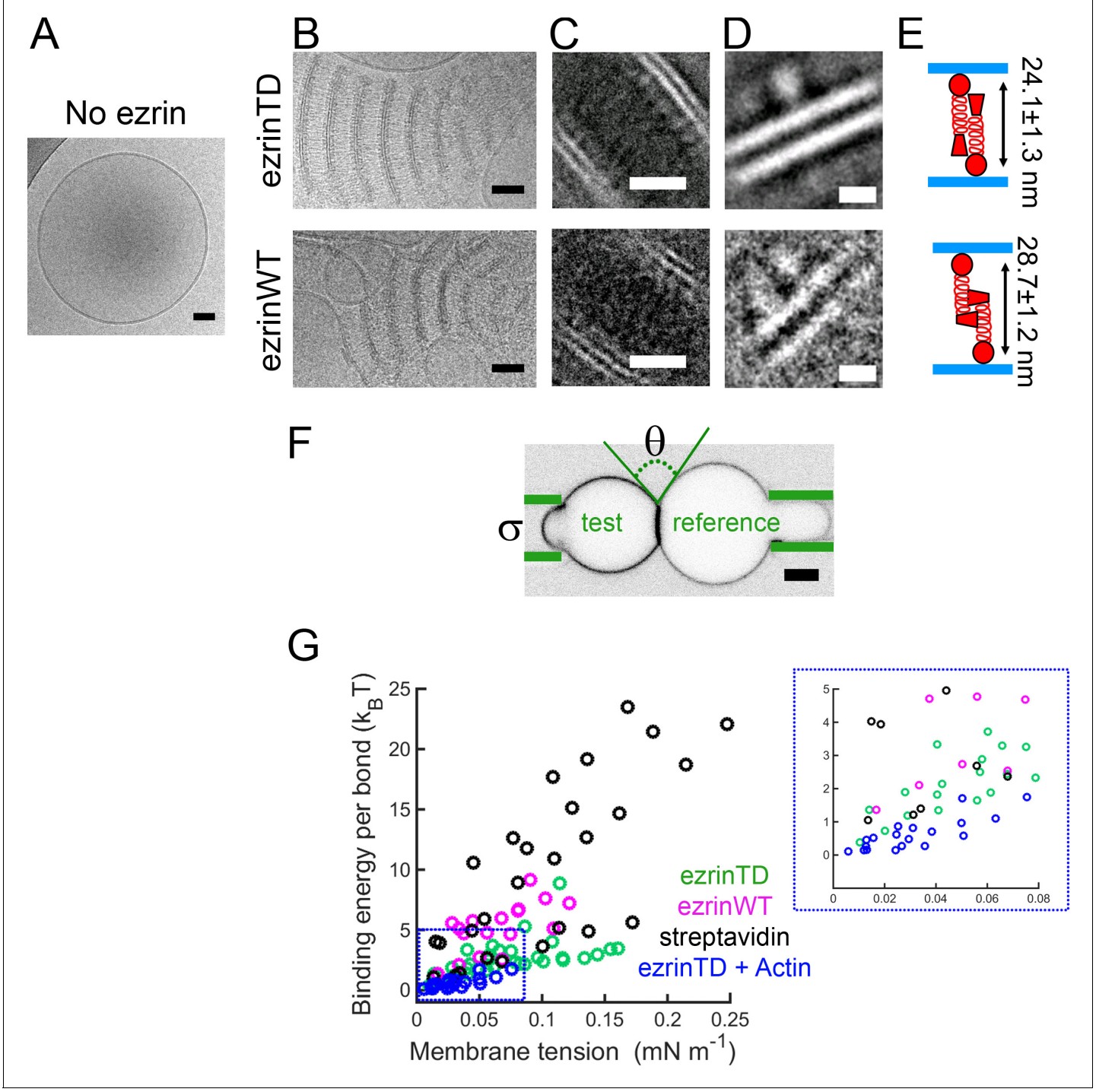

**Figure 1.** EzrinTD and ezrinWT have distinct conformations on PIP$_2$-containing membranes. (A) LUV in the absence of ezrin. Scale bar, 50 nm. (B) Representative cryo-electron micrographs of PIP$_2$-membranes tethered by ezrinTD (0.2 µM) or ezrinWT (1.2 µM). Scale bars: 50 nm. (C–D) Representative two-dimensional class averages of ezrinTD- and ezrinWT-tethered membranes. Scale bars: (C) 20 nm and (D) 5 nm. (E) Cartoons illustrating ezrin-membrane tethering by ezrinTD and ezrinWT. The distances deduced from class averages as shown in (C) and (D) between the globular domains of ezrinTD and ezrinWT between two tethered membranes are 24.1 ± 1.3 nm and 28.7 ± 1.2 nm (averaged from 10 classes, mean ± standard deviation), respectively (see analysis details in the Materials and methods). The size of the globular domains of ezrinTD and ezrinWT shown in (D) are 5.4 ± 0.6 nm long by 3.5 ± 0.5 nm wide for ezrinTD (averaged from 21 classes) and 5.3 ± 0.5 nm long by 2.9 ± 0.3 nm wide for ezrinWT (averaged from 7 classes, mean ± standard deviation, see analysis details in the Materials and methods). (F) Representative confocal image of the test GUV and the reference GUV tethered by ezrinTD (inverted grayscale) in the dual micropipette assay. The green lines indicate the micropipettes. Scale bar, 5 µm. (G) Membrane tethering strengths of ezrinTD (N = 8 GUVs, n = 4 sample preparations), ezrinWT (N = 4 GUVs, n = 3 sample preparations),

*Figure 1 continued on next page*

*Figure 1 continued*

streptavidin bonds (N = 5 GUVs, n = 2 sample preparations) and ezrinTD in the presence of F-actin (N = 10 GUVs, n = 3 sample preparations). Inset: enlarged region indicated by the blue dotted box in (**G**).

DOI: https://doi.org/10.7554/eLife.37262.002

The following source data and figure supplements are available for figure 1:

**Figure supplement 1.** Analysis of purified recombinant ezrinTD and ezrinWT, and their binding to PIP$_2$-containing membranes.
DOI: https://doi.org/10.7554/eLife.37262.003
**Figure supplement 1—source data 1.** Source data of *Figure 1—figure supplement 1C and D*.
DOI: https://doi.org/10.7554/eLife.37262.004
**Figure supplement 1—source data 2.** Source data of *Figure 1—figure supplement 1G*.
DOI: https://doi.org/10.7554/eLife.37262.005
**Figure supplement 2.** Frequency and critical concentration of membrane tethering by ezrinTD and ezrinWT.
DOI: https://doi.org/10.7554/eLife.37262.006

aspiration pressure (*Evans and Needham, 1988*) (*Noppl-Simson and Needham, 1996*). We followed the previously established experimental procedure in which one GUV, named the reference GUV, was under high tension (>0.4 mN/m) to ensure a spherical shape while adjusting the membrane tension of the other GUV, named the test GUV (*Franke et al., 2006*) (*Figure 1F*). At low membrane tension (~0.01 mN/m) of the test GUV, we observed an enrichment of ezrin fluorescence signal in the contact zone while the test GUV had a lemon-like shape, indicating ezrin tethering (*Figure 1F* and *Figure 1—figure supplement 2C*). No membrane tethering was observed when ezrin was absent. Monitoring ezrin mobility with fluorescent recovery after photobleaching (FRAP) revealed that only ~10% of ezrinTD and ezrinWT diffuse freely in the tethering zone (*Figure 1—figure supplement 2D*), confirming the formation of a stable contact resulting from ezrin tethering, while both protein types are mobile in the non-tethered zone (*Figure 1—figure supplement 2E*). Note that we did not average FRAP curves due to technical limitations of our set-up and variability among GUVs. We deduced the tethering energy per ezrin pair bond between the two GUVs by using their contacting geometry at a given membrane tension ranging between 0.01 mN/m to 0.25 mN/m (see *Figure 1F* for the schematic methods for details) (*Franke et al., 2006*). We compared ezrin tethering energy to that of the streptavidin-biotin bond as a reference. For both the ezrin-ezrin bond and the biotin-streptavidin bond, the binding energy was dependent on the membrane tension (*Figure 1G*). This tension dependence may arise from the electrostatic repulsion of the opposing membranes, as suggested previously (*Franke et al., 2006*). At all tensions probed the tethering energy calculated for ezrinWT was on average higher than that of ezrinTD but lower than that of the biotin-streptavidin bond (*Figure 1G*) (*Noppl-Simson and Needham, 1996*). The higher tethering energy for ezrinWT is in apparent contradiction with the lower fraction of tethered vesicles detected with cryo-EM (*Figure 1—figure supplement 2A*); however, this lower fraction of tethered vesicles is likely due to the much lower vesicle membrane coverage by ezrinWT as compared to ezrinTD at the same bulk concentration (*Figure 1—figure supplement 1G*). In conclusion, when associated with membranes, ezrinWT and ezrinTD display different conformations leading to different strengths of the intermolecular interactions, and hence to different tethering energies.

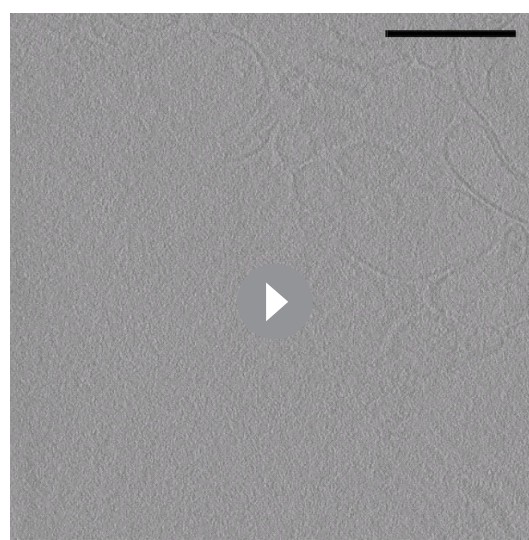

**Video 1.** Cryo-tomography of ezrinTD tethered membrane stacks. Green colored structures represent tethered membrane stacks and magenta colored structures represent vesicular structures. Scale bar: 200 nm.
DOI: https://doi.org/10.7554/eLife.37262.007

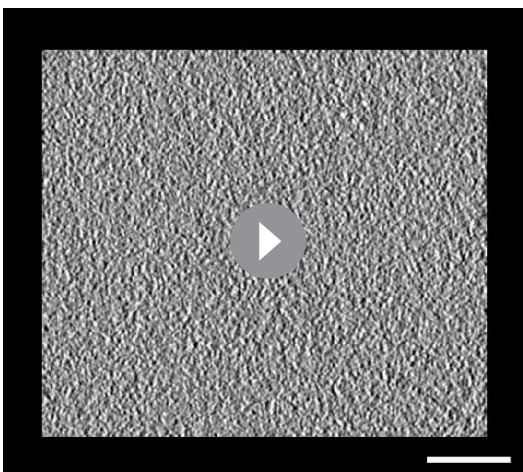

**Video 2.** Cryo-tomography of ezrinTD tethered membrane stacks. Membranes are highlighted in yellow. Scale bar: 840 nm.
DOI: https://doi.org/10.7554/eLife.37262.008

## F-actin binds to a brush-like ezrinTD assembly and reduces its membrane tethering activity

Extensive experimental evidence points out that the C-ERMAD of ERM proteins has to be phosphorylated for their association with F-actin (*Fievet et al., 2004*) (*Fritzsche et al., 2014*) (*Nakamura et al., 1999*). The fact that ezrinWT and ezrinTD display different conformations on membranes motivated us to examine whether ezrinTD and ezrinWT bridge F-actin to $PIP_2$-containing membranes. We first assessed actin recruitment on GUVs. When GUV membranes were covered with AX546-labeled ezrinTD, the recruitment of AX488 phalloidin-labeled muscle or non-muscle F-actin to the membranes was clearly visible, and F-actin formed a dense, network-like structure on the GUVs (*Figure 2A–2D*). We observed the recruitment of F-actin at ezrinTD bulk concentrations down to 50 nM (or down to 0.5% membrane coverage). In contrast, F-actin was not detectable on ezrinWT-coated GUVs, for ezrinWT bulk concentrations ranging from 0.5 µM to 2 µM, while F-actin was clearly visible in the background as in the control experiment where there was no ezrin on GUVs (*Figure 2E–H*). We only observed the recruitment of muscle F-actin by ezrinWT at bulk concentration above 5 µM (or equivalently, at more than 10% membrane coverage). Our observation seems inconsistent with a previous study using SLBs on silicon substrates where F-actin recruitment on ezrinWT decorated $PIP_2$-membranes was observed (*Bosk et al., 2011*). However, in Bosk *et al.*, it was reported that ezrinWT formed large immobilized clusters on SLBs even at low $PIP_2$ density that might be attributed to ezrinWT activation. In contrast, we never observed clusters of ezrinTD or ezrinWT on free-standing GUV membranes. Moreover, a strong enhancement for F-actin recruitment was also observed by Bosk *et al.* in the case of ezrinTD as compared to ezrinWT. Here, our observations clearly demonstrate that $PIP_2$ is not sufficient for the recruitment of F-actin by ezrinWT at the surface of GUVs.

To resolve the membrane-ezrinTD-actin organization at nanometric resolution, we carried out cryo-EM experiments using LUVs. In the absence of ezrinTD, no interaction between LUVs and muscle F-actin was observed (*Figure 2I*). In the presence of ezrinTD, we observed that actin filaments were parallel to the membrane surface, following the contour of the membrane, (*Figure 2J and K* arrowheads) with a brush-like assembly of ezrinTD bridging F-actin and the membrane (*Figure 2K and L* brackets). The distance between actin filaments and the membrane is about 25 nm, similar to the length of the open ezrinTD molecule (*Figure 1E*). Moreover, small vesicles were also found in contact with the ezrinTD brush (*Figure 2J and K* arrows). Our results thus confirm that a negative charge at position T567 of C-ERMAD, that is mimicking ezrin phosphorylation, mediates a conformational change of ezrin that then makes ezrin amenable to link F-actin to membranes. Moreover, we reveal that on $PIP_2$-containing membranes ezrinTD sits perpendicularly to the membrane with its FERM domain interacting with $PIP_2$ and its phosphomimetic C-ERMAD domain protruding from the membrane to engage interactions with actin filaments (*Figure 2M*). This is in contrast to the previous hypothesis suggesting a parallel configuration of ezrin dimers between membrane and F-actin (*Jayasundar et al., 2012*).

We next addressed the effect of the presence of F-actin on ezrinTD-mediated membrane tethering. We measured the binding energy of GUVs coated with ezrinTD in the presence of muscle F-actin using the dual micropipette tethering method. We observed that actin significantly reduces tethering strength of ezrinTD (*Figure 1G*). This indicates actin filaments inhibit the ability of ezrinTD to tether membranes.

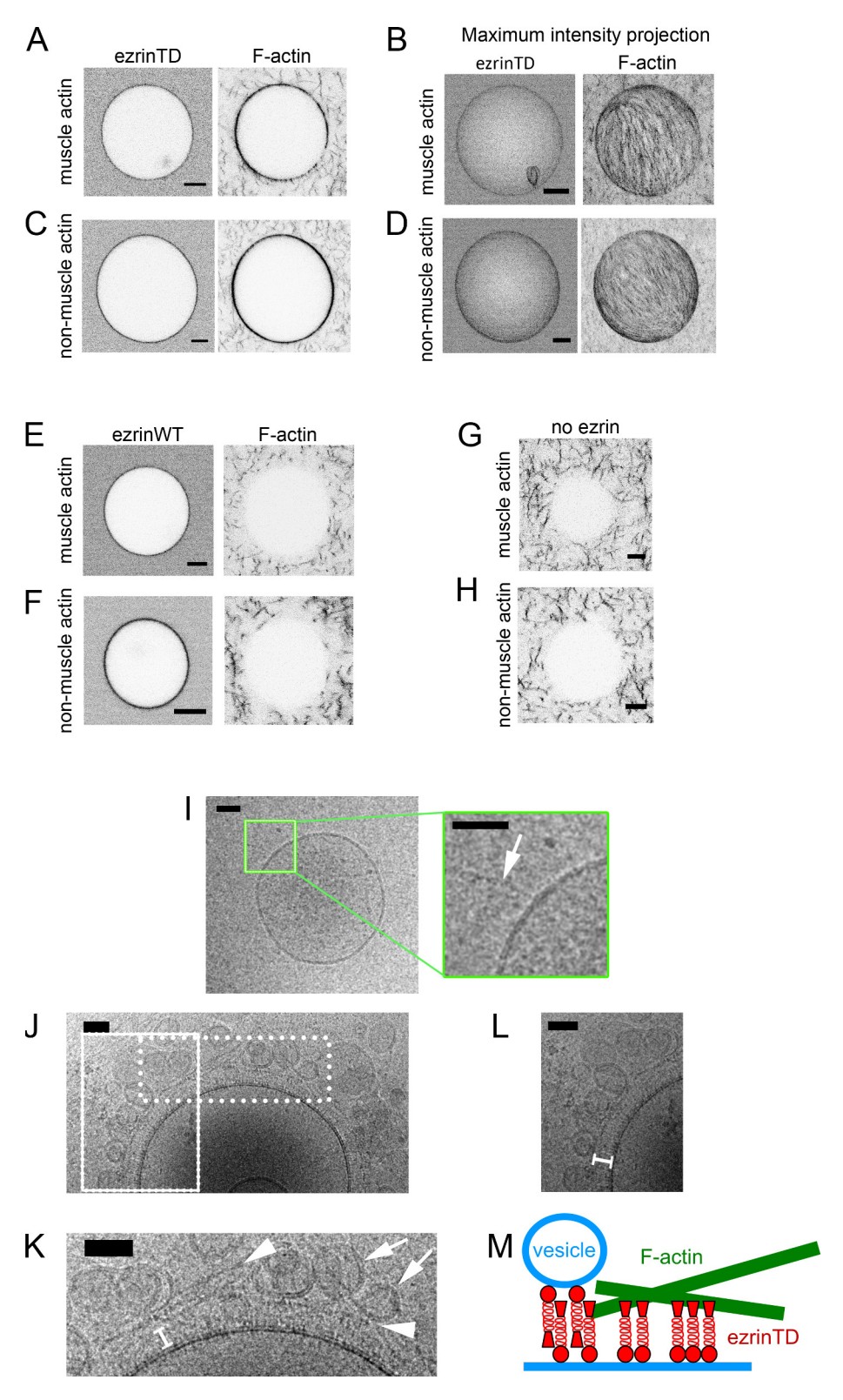

**Figure 2.** EzrinTD forming a brush-like structure bridges F-actin and the membrane. (A–H) Representative confocal images of GUVs coated with ezrinTD (A–D), ezrinWT (E and F) or in the absence of ezrin (G and H), in the presence of muscle (A, B, E and G) or non-muscle (C, D, F and H) F-actin. (B) and (D) Maximum intensity projections of (A) and (C), respectively. The bulk concentrations of muscle and non-muscle F-actin are 0.4 µM and 0.6 µM, respectively, and the bulk concentrations of ezrinTD and ezrinWT are both 0.5 µM. Scale bars, 5 µm. Inverted grayscale images were shown. (I–L)

*Figure 2 continued on next page*

*Figure 2 continued*

Representative cryo-electron micrographs of PIP$_2$-containing LUVs incubated with muscle F-actin in the absence (**I**) and presence (**J–L**) of ezrinTD. Inset in (**I**): enlargement of the region marked by the green rectangle. Arrow indicates F-actin. Scale bars, 50 nm. (**K**) and (**L**) Enlargements of the regions marked by the dotted and solid line rectangle in (**J**), respectively. In (**K**), arrowheads indicate F-actin and arrows indicate tethered vesicles. In (**K**) and (**L**) white brackets indicate brush-like ezrinTD assembly. EzrinTD and F-actin concentrations are 0.3 μM and 1 μM, respectively. Scale bars, 50 nm. (**M**) Cartoon illustrating brush-like ezrinTD assembly recruiting F-actin and tethering vesicles to membranes.
DOI: https://doi.org/10.7554/eLife.37262.009

## Phosphomimetic mutation enables the positive membrane curvature sensing of ezrin

Since ezrinWT and ezrinTD have different conformations when associated with quasi-flat PIP$_2$-containing LUVs and GUVs, we next investigated how they interact with highly curved galactocerebroside nanotubes (**Dang et al., 2005**). These rigid nanotubes with a uniform external diameter of 25 nm were composed of galactocerebrosides supplemented with 15 mole% EPC and 5 mole% PIP$_2$. Thus these nanotubes offer a suitable approach to probe positive curvature sensing of proteins by cryo-EM. They were dispersed in solution in the absence of protein (**Figure 3—figure supplement 1A**). Addition of ezrinWT or ezrinTD promoted different types of nanotube assemblies. In the presence of ezrinTD, 68.5% of the tubes formed disordered assemblies (henceforth named disordered) and 25% were isolated nanotubes (henceforth named isolated) (**Figure 3A and C**). In the presence of ezrinWT, we observed up to 42% of isolated nanotubes and instead of a disordered organization, 58% of the tubes formed parallel stacks (henceforth named stacked) (**Figure 3B and C**). These stacks have a regular spacing of 25.8 ± 2.4 nm between the centers of the opposing FERM domains, similar to the distance between the tethered bilayers observed in the LUV experiments (**Figure 1E**). The difference in the nanotube assemblies between ezrinWT and ezrinTD suggests a difference in the interaction and organization of individual molecules with the nanotubes. In the presence of ezrinTD all isolated tubes (total 15 tubes) were covered with proteins. In contrast, in the presence of ezrinWT, only 55% (total 26 tubes) of the isolated tubes were decorated with proteins, showing a weaker affinity for positively curved membranes compared to that of ezrinTD. Altogether, these observations indicate that ezrinTD and ezrinWT can tether membranes with strong positive curvature but they induce different types of organization due to their different conformations.

The different conformations observed between ezrinWT and ezrinTD on rigid galactocerebroside nanotubes suggest a different ability to sense positive membrane curvature. We thus quantified the positive curvature sensing ability of ezrin by measuring the enrichment of ezrinWT and ezrinTD on positively curved membranes relative to flat ones. To this end, we performed tube pulling experiments where a fluid membrane nanotube was pulled outwards from a micropipette-held GUV while injecting ezrin next to the GUV with another micropipette, as previously done with some curvature sensing proteins (**Sorre et al., 2012**). At a given membrane tension, after injecting AX488 labeled ezrinWT or ezrinTD for 2–3 mins, the membrane coverage of ezrin on GUVs generally plateaued (**Figure 3—figure supplement 1B**). By controlling the aspiration pressure, the membrane tension of a GUV can be adjusted. In the absence of protein binding, the radius of the nanotube $R$ is set by the membrane tension $\sigma$ and the membrane bending rigidity $\kappa$ of the GUV, $R = \sqrt{\frac{\kappa}{2\sigma}}$ (**Derényi et al., 2002**). We therefore have a direct control of the nanotube radius, typically ranging from 10 nm to 100 nm. By calculating the ratio of lipid fluorescence intensities in the tube and in the GUV, we can derive the tube radius (see Materials and methods for details). Our experimental system allows us to assess curvature sensing by measuring the ratio of ezrin fluorescence intensities on the tube (positively curved membranes) and on the GUV (flat membranes), normalized by the ratio of lipid fluorescence intensities on both structures to calculate the sorting ratio $S$, $S = \frac{(I_{tube}/I_{vesicle})_{ezrin}}{(I_{tube}/I_{vesicle})_{lipid}}$ (**Sorre et al., 2012**). For experiments where the fluorescence signals of ezrin on tubes were too low to be measured (close to the noise level), we set $S$ equal to 0. We first checked that in the absence of ezrin, no sorting of PIP$_2$ in membrane tubes was observed (**Figure 3D**), validating that protein sorting will result from the property of the protein only. No binding on tubes was detected ($S = 0$) for ezrinWT, in a relatively large fraction of experiments; when detectable, ezrinWT was not enriched on tubes as demonstrated by $S \cong 1$ (**Figure 3E**). In contrast, binding and enrichment on tubes was detected for ezrinTD. We grouped the sorting data into three classes corresponding to different membrane

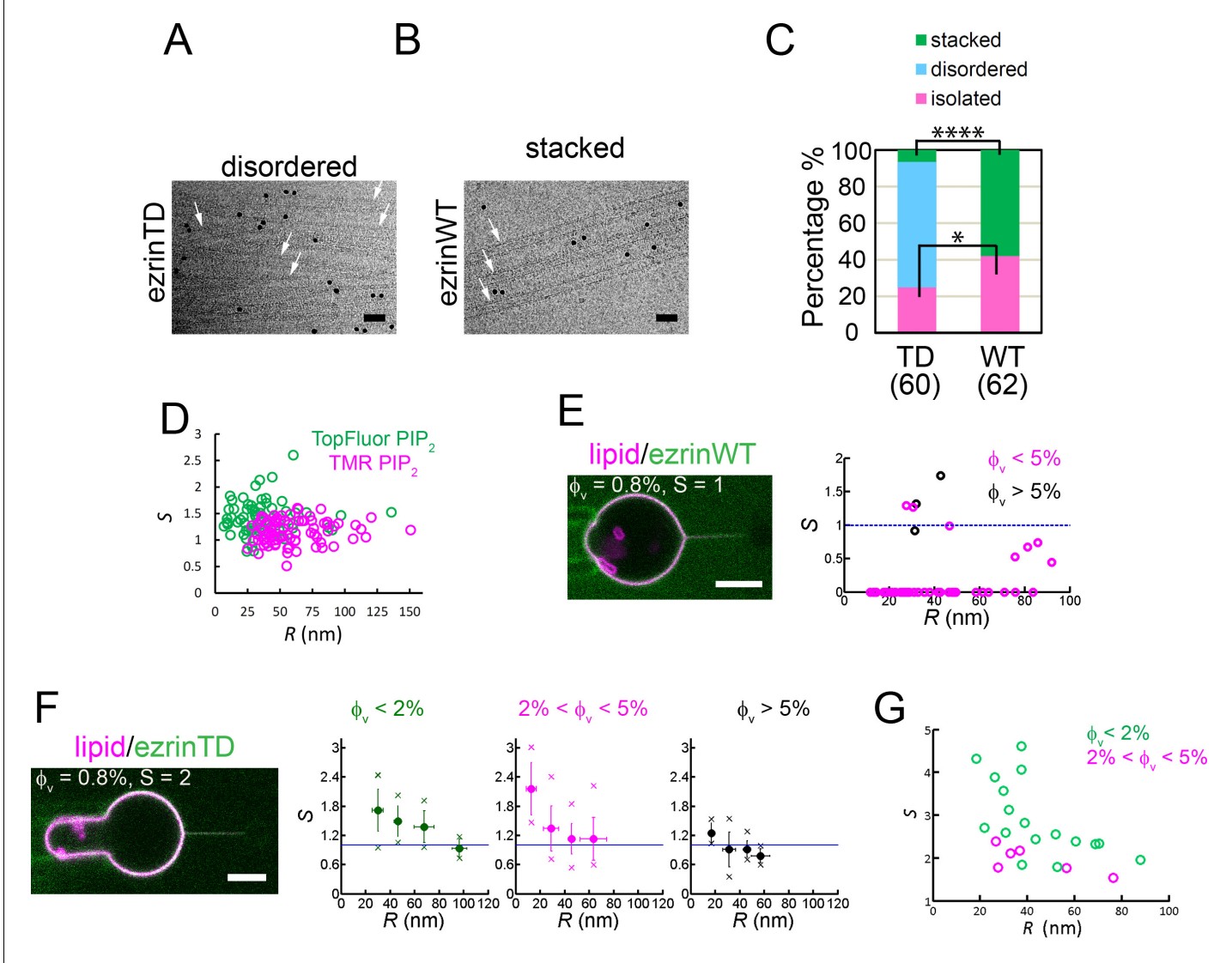

**Figure 3.** EzrinTD binds to positively curved membranes. (**A and B**) Rigid lipid nanotubes assembled by ezrin: representative cryo-electron micrographs of ezrinTD inducing randomly-oriented nanotube assembly, named disordered (**A**) and ezrinWT inducing parallel stacks, named stacked (**B**). Arrows indicate some of the nanotubes in both cases. Scale bars: 50 nm. The black dots are gold particles. Protein concentrations: 1.2 μM for both ezrinTD and ezrinWT. Nanotube concentration: 0.1 g.L$^{-1}$. (**C**) Percentages of membrane nanotubes being disordered, forming stacks or being isolated in the presence of ezrinTD or ezrinWT, deduced from cryo-EM. Numbers of tubes measured are N = 60 and N = 62 from two distinct sample preparations, for ezrinTD and ezrinWT, respectively. Statistic test (chi-square test): $p \approx 1.5 \times 10^{-9}$ for stacked tubes (****) and $p = 0.0477$ for isolated tubes (*). (**D**) Absence of curvature-induced sorting of PIP$_2$ measured using two types of fluorescently-labeled PIP$_2$ (TopFluor PIP$_2$ (N = 14 GUVs, n = 1 sample preparation) and TMR PIP$_2$ (N = 23 GUVs, n = 2 sample preparations)). $S$: sorting ratio. $R$: membrane tube radius. (**E**) and (**F**) Nanotube pulling assay to probe ezrin positive membrane curvature sensing. (Left) Representative confocal images of ezrinWT (**E**) and ezrinTD (**F**) binding on a membrane nanotube pulled from a GUV aspirated in a micropipette. The nanotube is held by a bead (not fluorescent) that is trapped by optical tweezers. Scale bars, 5 μm. (Right) Sorting ratio, $S$, as a function of tube radius, $R$, for ezrinWT (**E**) and ezrinTD (**F**). φ$_V$: ezrin membrane surface fraction. Blue lines indicate $S = 1$. In (**E**), measurements were collected from N = 13 GUVs, n > 3 independent sample preparations. In (**F**), measurements were collected from N = 63 GUVs, n > 3 independent sample preparations. Error bars indicate standard deviations, circles are mean and X symbols are maximum and minimum for each condition, excluding $S = 0$ data. (**G**) Sorting of the FERM domain at different membrane tube radii. φ$_V$: membrane surface fraction of the FERM domain on the GUVs. $S$: sorting ratio. $R$: membrane tube radius. Measurements were collected from N = 5 GUVs, n = 2 sample preparations.
DOI: https://doi.org/10.7554/eLife.37262.010

The following figure supplement is available for figure 3:

**Figure supplement 1.** Binding of ezrin on GUVs and positive curvature-induced sorting of ezrinTD on membrane nanotubes.
DOI: https://doi.org/10.7554/eLife.37262.011

surface fractions of ezrinTD on the GUVs, $\Phi_v$, since curvature-induced sorting is expected to depend on $\Phi_v$ (*Sorre et al., 2012*) (*Zhu et al., 2012*) (*Prévost et al., 2015*). We observed that $S$ monotonically increased when the tube radius decreased and $S$ was larger than 1, showing that ezrinTD is a positive curvature sensor (*Figure 3F* and *Figure 3—figure supplement 1C*). As expected (*Sorre et al., 2012*) (*Zhu et al., 2012*) (*Prévost et al., 2015*), $S$ was lower for the highest $\Phi_v$ values (*Figure 3F* and *Figure 3—figure supplement 1C*). However, the positive curvature sensing ability of ezrinTD is weaker than previously reported curvature-sensing proteins (*Sorre et al., 2012*) (*Zhu et al., 2012*) (*Prévost et al., 2015*) as $S$ is lower than 3 at low protein membrane coverage. We then examined if the FERM domain of ezrin senses positive membrane curvature by performing the same tube pulling experiments as for ezrin. The enrichment of the FERM domain on positively curved membranes was clearly observed and even stronger than that of ezrinTD, as sorting values up to four were observed (*Figure 3G*). Thus, our results demonstrate that the conformation of ezrinTD allows its FERM domain to sense positive membrane curvature.

## Neither ezrinTD nor ezrinWT sense negative membrane curvature

We next wondered how ezrin is enriched into cellular protrusions wherein membranes have a negative mean curvature. To this end, we pulled membrane nanotubes from GUVs encapsulating ezrin and measured the sorting ratio $S$, as described earlier (*Prévost et al., 2015*). We observed that ezrin was present on the external leaflet during GUV preparation and could be effectively detached at high salt concentration (300 mM) (*Figure 4—figure supplement 1*). We set $S = 0$ for experiments where the fluorescence signals of ezrin were too weak for detection. For both ezrinTD and ezrinWT, we observed no clear dependence of the sorting ratio $S$ on tube radius within the experimental accessible tube radii. At low protein coverage on the GUVs (5%) the sorting ratio $S$ fluctuates around 1, indicating the absence of ezrinTD or ezrinWT enrichment in tubes (*Figure 4*). Our results therefore demonstrate that ezrin is not a negative membrane curvature sensor.

## EzrinWT and ezrinTD are enriched in negatively curved membranes by interacting with I-BAR domain proteins

Since ezrin has no intrinsic affinity for negatively curved membranes although it shows a cellular enrichment on negatively curved membranes, we investigated whether ezrin is recruited through partner proteins localize to cellular protrusions. IRSp53, an I-BAR domain containing protein, was found to colocalize with ezrin in microvilli (*Garbett et al., 2013*). Moreover, IRSp53 is involved in the initiation of filopodia (*Disanza et al., 2013*), and is enriched *in vitro* inside model membrane nanotubes (*Prévost et al., 2015*). Therefore, IRSp53 is a potential candidate to recruit ezrin to membrane protrusions. Indeed, using super-resolution structured illumination microscopy, we observed that patchy fluorescent signals of GFP-IRSp53 or its GFP-I-BAR domain overlapped partially with the endogenous ezrin in cellular protrusions of LLC-PK1 epithelial cells (*Figure 5A and B*). We confirmed the association of ezrin with IRSp53 or with its I-BAR domain by co-immunoprecipitating the endogenous ezrin with the GFP-IRSp53 or GFP-I-BAR domain in HeLa cells (*Figure 5C and D*). GFP-IRSp53 and GFP-I-BAR immunoprecipitated ezrin to the same extent (*Figure 5D*) regarding their expression level shown in *Figure 5C*. To assess whether ezrin and the I-BAR domain of IRSp53 interact directly, we performed a GUV-binding assay. We generated GUVs containing DOPC and DOGS-Ni-NTA lipids, without PIP$_2$ (Ni-GUVs). In the absence of the I-BAR domain, ezrin does not bind to Ni-GUVs (*Figure 5E*). We then incubated Ni-GUVs first with His-tagged I-BAR domain and then with ezrinWT or ezrinTD. In the presence of the His-tagged I-BAR domain, bound to Ni-GUVs via the His/Ni-NTA interaction, we found both ezrinTD and ezrinWT binding on Ni-GUVs (*Figure 5F*). As a control, we checked that ezrin does not bind to Ni-GUVs covered with his-GFP (*Figure 5G*). These observations demonstrate a direct interaction between the I-BAR domain and ezrinWT and ezrinTD.

We next examined whether ezrin is enriched on negatively curved membranes in the presence of the I-BAR domain of IRSp53. Encapsulation of two proteins in GUVs is very challenging, thus we took advantage of the spontaneous GUV membrane tubulation induced by the I-BAR domain (*Saarikangas et al., 2009*) (*Mattila et al., 2007*) (*Barooji et al., 2016*). Incubation of PIP$_2$-containing GUVs with the I-BAR domain of IRSp53 led to membrane tubular invaginations towards the interior of the GUVs (*Figure 6—figure supplement 1A*), as shown previously (*Saarikangas et al., 2009*) (*Mattila et al., 2007*) (*Barooji et al., 2016*). To validate our assay, we quantified the relative

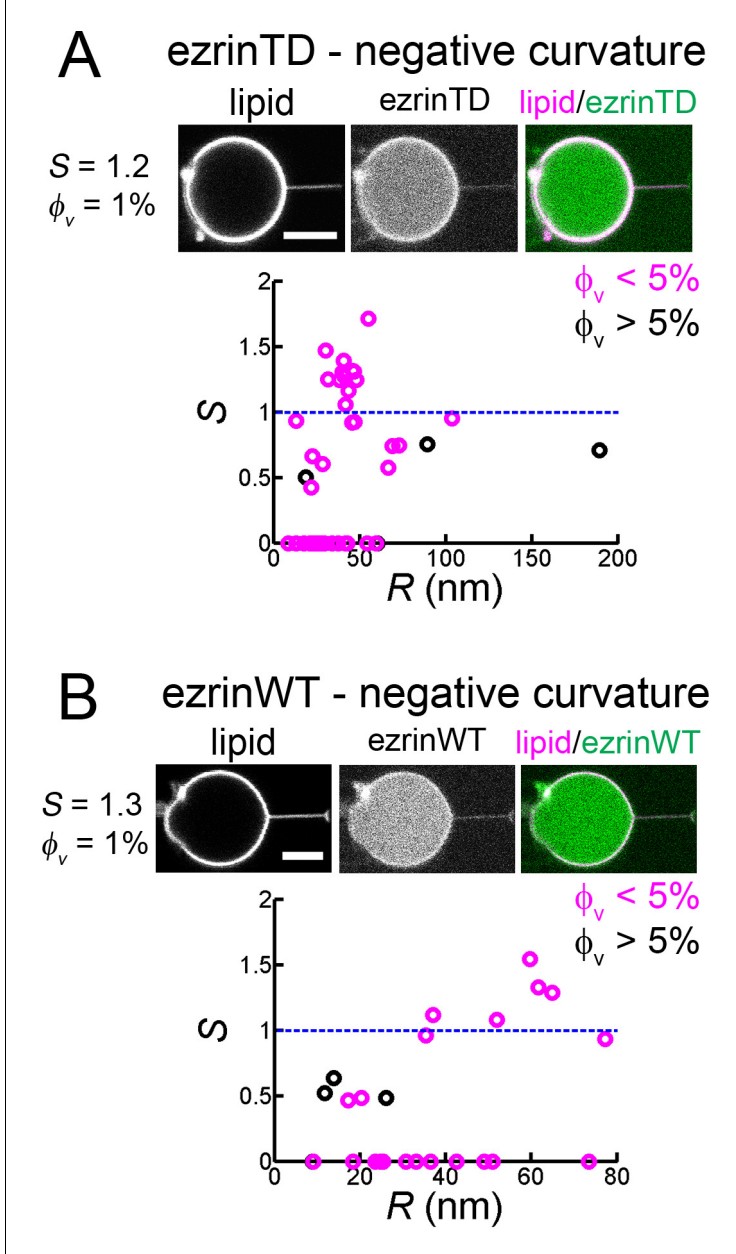

**Figure 4.** Both ezrinTD and ezrinWT do not sense negative membrane curvature. (**A**) and (**B**) Tube pulling experiments from GUVs encapsulating ezrinTD (**A**) or ezrinWT (**B**). (Top) Representative confocal images of ezrin sorting experiments, corresponding to (**A**) a sorting ratio $S$ = 1.2 at $\varphi_v$ =1% and (**B**) $S$ = 1.3 at $\varphi_v$=1%, respectively. Scale bars, 5 µm. (Bottom) Sorting ratio, $S$, as a function of membrane tube radius $R$ and for different ezrin surface fractions on the GUVs, $\varphi_v$. Measurements were collected from N = 26 GUVs (n > 3 sample preparations), and N = 7 GUVs (n = 3 sample preparations), for ezrinTD and ezrinWT, respectively. Blue lines indicate $S$ = 1.

DOI: https://doi.org/10.7554/eLife.37262.012

The following source data and figure supplements are available for figure 4:

**Figure supplement 1.** Salt-induced desorption of ezrinTD and ezrinWT bound to the outer leaflet of GUVs.
DOI: https://doi.org/10.7554/eLife.37262.013

**Figure supplement 1—source data 1.** Source data of *Figure 4—figure supplement 1A and B*.
DOI: https://doi.org/10.7554/eLife.37262.014

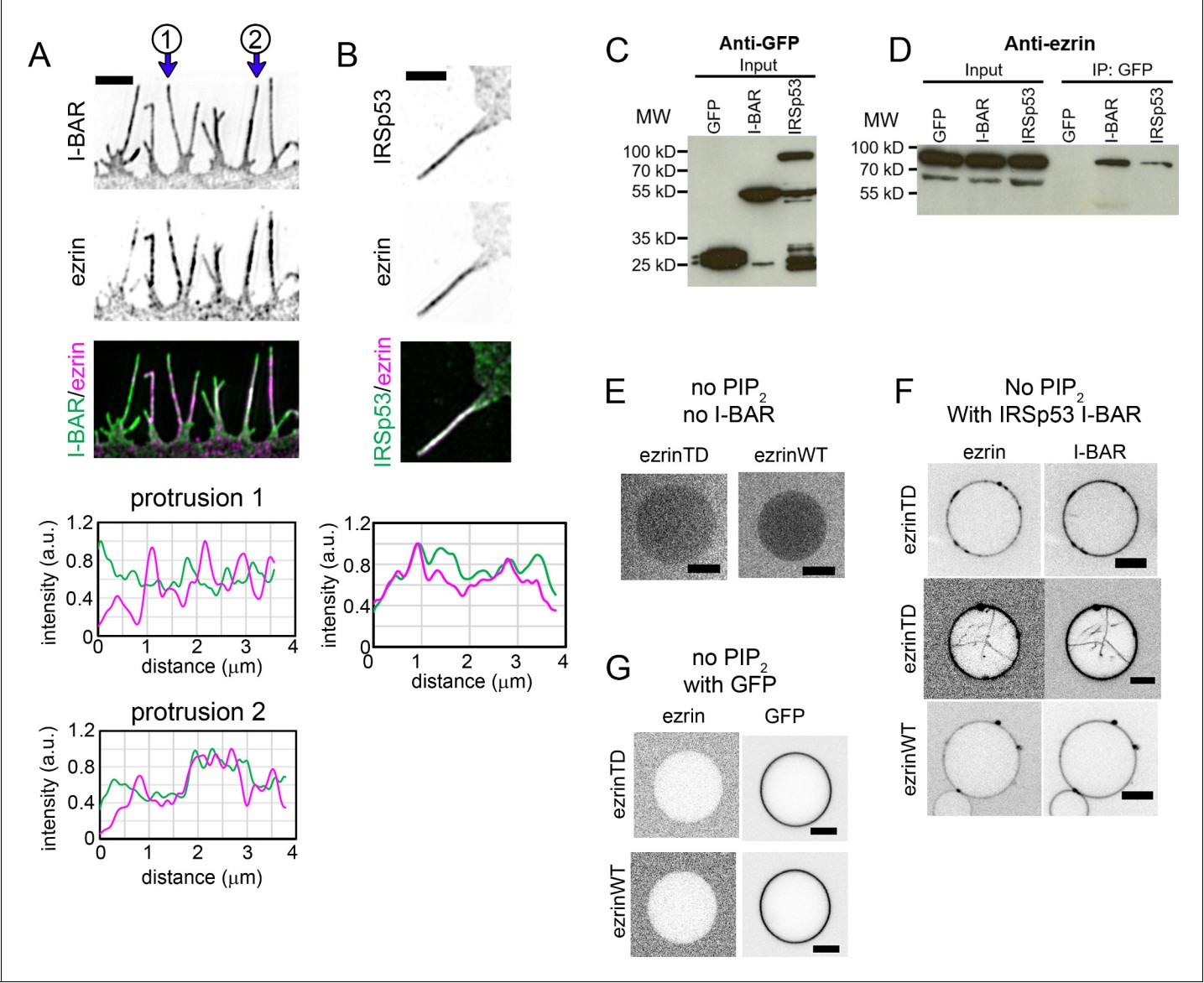

**Figure 5.** Ezrin partially colocalizes with IRSp53 in cellular protrusions and directly interacts with the I-BAR domain of IRSp53. (**A**) and (**B**) (Top) Representative structured illumination microscopy images of cellular protrusions of LLC-PK1 cells transfected with plasmids encoding GFP-I-BAR domain (**A**) or GFP-IRSp53 (**B**), and immunolabeled for endogenous ezrin. Scale bars, 2 µm. Inverted grayscale images are shown, unless color codes are indicated in the figure. (Bottom) Normalized fluorescence intensity profiles of protrusions indicated in (**A**) and the protrusion in (**B**). The distance zero is at the tip of the protrusions and the distance goes along the protrusions until the base of the protrusions at the cell edge. (**C**) and (**D**) GFP, GFP-IRSp53 or GFP-I-BAR domain transfected HeLa cells were lysed and immunoprecipitated (IP) using GFP-trap. Cell lysates (input) and IP were analyzed by Western blot for the expression of GFP, GFP-I-BAR domain and GFP-IRSp53 using an anti-GFP antibody (**C**) and pull down of endogenous ezrin using an anti-ezrin antibody (**D**). (**E–G**) Representative confocal images of ezrinTD or ezrinWT binding to Ni-GUVs in the absence of both PIP$_2$ and the I-BAR domain (**E**), in the absence of PIP$_2$ but in the presence of the I-BAR domain of IRSp53 (**F**), or in the absence of PIP$_2$ but in the presence of GFP (**G**). I-BAR domain occasionally induces tubules in the absence of PIP$_2$ and ezrinTD is recruited in these I-BAR domain-induced tubules, as shown in (**F**). Protein bulk concentrations: for ezrinTD and ezrinWT, 1 µM, and for the I-BAR domain, 2 µM. Scale bars, 5 µm. Inverted grayscale images were shown.
DOI: https://doi.org/10.7554/eLife.37262.015

distribution of the I-BAR domain on GUVs and on the corresponding tubules by measuring the sorting ratio $S$ of the I-BAR domain. In this tubulation assay, we cannot use any calibration method based on lipid fluorescence intensities to determine the tube radii. We thus used the membrane fluorescence ratio on the tube and on the corresponding GUV, $(I_{tube}/I_{vesicle})_{membrane}$, as a relative measurement of the tube radius. We observed a non-monotonic enrichment of the I-BAR domain

depending on the membrane fluorescence ratio $(I_{tube}/I_{vesicle})_{membrane}$ (**Figure 6—figure supplement 1B**), in close agreement with a previous report using the tube pulling assay (**Prévost et al., 2015**). Consistent with the previous report by Chen *et al.* (**Chen et al., 2015**), we observed I-BAR-induced tube formation at low I-BAR domain membrane coverage on GUVs (<2%). Moreover, in agreement with the previous study by Prévost et al. (**Prévost et al., 2015**), we observed a stronger sorting of the I-BAR domain at lower I-BAR domain membrane coverage on GUVs (<2%) than at higher membrane coverage (>5%) (**Figure 6—figure supplement 1B**). By comparing our results with those obtained by Prévost *et al.* (**Prévost et al., 2015**), we could estimate tube radii considering that the maximum sorting of the I-BAR domain at $(I_{tube}/I_{vesicle})_{membrane} \cong 0.3$ should correspond to the intrinsic spontaneous radius of the I-BAR domain, $R \approx 20nm$ (**Figure 6—figure supplement 1B**).

To assess if ezrin is enriched in the tubules induced by the I-BAR domain of IRSp53, GUVs were first incubated with ezrin followed by the addition of the I-BAR domain. We observed that the I-BAR domain deformed ezrin-coated membranes and that ezrinTD or ezrinWT was present on I-BAR domain-induced tubules (**Figure 6—figure supplement 1C** and **Figure 6—figure supplement 1D**). The stronger fluorescence signal of ezrinTD or ezrinWT on I-BAR domain-induced tubules, as compared to those on GUVs, clearly indicates the enrichment of ezrin on the negatively curved tubular membranes (**Figure 6A and B**). Moreover, we observed a strong sorting of ezrinTD with a sorting ratio $S$ up to 8 at the lowest ezrinTD membrane coverage (<2%), which decreased when $\varphi_v$ increased(**Figure 6C**). The maximum sorting of ezrinTD at a membrane fluorescence ratio $\approx 0.4$ corresponds to a tube radius of about 30 nm (**Figure 6C**). Similarly, ezrinWT also enriched in I-BAR domain-induced tubules (**Figure 6D**).

It has been shown that BAR-domain proteins, including I-BAR, induce the formation of stable PIP$_2$ clusters (**Saarikangas et al., 2009**) (**Zhao et al., 2013**) that can recruit downstream partners (**Picas et al., 2014**). It is thus conceivable that local PIP$_2$ clusters induced by IRSp53 I-BAR domain facilitate ezrin enrichment. Indeed, we observed PIP$_2$ enrichment in the I-BAR domain induced tubules, with a sorting ratio $S$ up to 3 (**Figure 6—figure supplement 1E**). As a control, we observed no sorting induced by the I-BAR domain for another fluorescent lipid GM1* (BODIPY-FL C5-ganglioside GM1) (**Figure 6—figure supplement 1E**). Notably, this PIP$_2$ enrichment is weaker than the enrichment of ezrinTD with $S$ up to 8 (**Figure 6C**) (given the tubule radii were set by the curvature of the I-BAR domain (**Prévost et al., 2015**), to compare PIP$_2$ sorting and ezrin sorting at $\phi_v < 2\%$ , we performed *t*-test and obtained $p = 1.2 \times 10^{-17}$). This observation thus shows that the direct interaction between ezrin and the I-BAR domain enhances ezrin enrichment in I-BAR domain induced tubules.

MIM and ABBA, two other I-BAR domain proteins, were shown to colocalize with ezrin at the edge of transendothelial cell tunnels, which have a high negative curvature of about 1/30 nm (**Stefani et al., 2017**). Similarly to our measurements with the I-BAR domain of IRSp53, we found that ezrinTD interacts with ABBA I-BAR domain directly (**Figure 6E**) and ezrinTD was enriched in ABBA I-BAR domain-induced membrane tubules, with $S>1$ for $\phi_v <2\%$ (**Figure 6F and G**). Therefore, our data indicates that the enrichment of ezrinTD via ABBA I-BAR domain accounts for ezrinTD enrichment at transendothelial cell tunnels.

Taken together, our results evidence that ezrin enrichment in negatively curved membranes requires curvature-sensitive partners such as I-BAR domain proteins.

## Discussion

In cells, ezrin is associated with actin-rich membrane protrusions where the membranes have negative curvature, with intracellular vesicles where the membranes have positive curvature, and with the cell cortex where the membrane is flat. Using biomimetic model membranes having different curvatures combined with cell biology approaches, we report here the enrichment of ezrin on curved/tubular membranes via mechanisms involving the specific conformation of ezrin for positive curvature and the binding of ezrin to I-BAR domains for negative curvature.

### Ezrin forms anti-parallel assemblies that tether PIP$_2$-containing membranes

It has been proposed that in solution, both ezrinWT and ezrinTD form anti-parallel homodimers via the association of the FERM domain of one ezrin monomer and of the C-ERMAD of the other

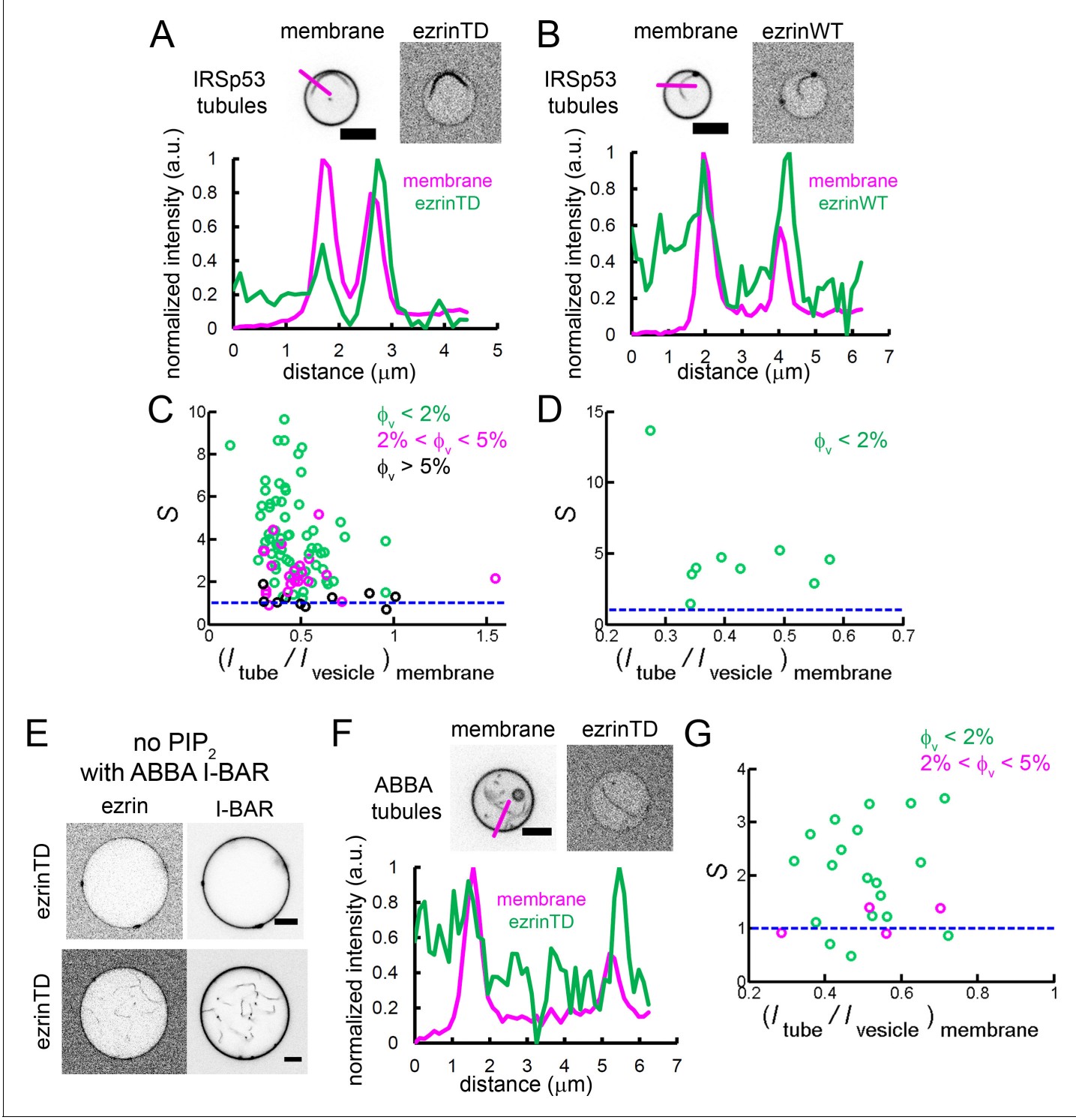

**Figure 6.** Enrichment of ezrinTD and ezrinWT in I-BAR domain induced tubules. (A) and (B) (Top) Representative confocal images of ezrinTD (A), and ezrinWT (B) in IRSp53 I-BAR domain induced tubules. Scale bars, 5 µm. (Bottom) Normalized fluorescence intensity profiles along the line drawn from outside the GUV towards the interior of the GUV, as indicated in the top image. Inverted grayscale images are shown. (C) and (D) Sorting of ezrinTD (C) and ezrinWT (D) in IRSp53 I-BAR domain-induced membrane tubules at different membrane coverages of ezrin. $(I_{tube}/I_{vesicle})_{membrane} \cong 0.4$ corresponds to a tubule radius of about 30 nm. In (C), measurements were collected from N = 94 GUVs, n = 6 sample preparations. In (D), measurements were collected from N = 9 GUVs, n = 4 sample preparations. (E) Representative confocal image of ezrinTD binding to Ni-GUVs in the absence of $PIP_2$ but in the presence of the I-BAR domain of ABBA. Protein bulk concentrations: ezrinTD, 1 µM, and ABBA I-BAR domain, 2 µM. Scale bars, 5 µm. Inverted grayscale images are shown. (F) (Top) Representative confocal images of ezrinTD in ABBA I-BAR domain-induced tubules. Inverted grayscale images

*Figure 6 continued on next page*

*Figure 6 continued*

were shown. (Bottom) Normalized fluorescence intensity profiles along the line drawn from outside the GUV towards the interior of the GUV, as indicated in the top image. (**G**) Sorting of ezrinTD in ABBA I-BAR domain-induced tubules at two different membrane coverages of ezrinTD. Measurements were collected from N = 19 GUVs, n = 3 sample preparations. Scale bars, 5 μm. $\varphi_V$: surface fraction of ezrinTD and ezrinWT on the GUVs. In (**C, D and G**), blue lines indicate $S$ = 1.

DOI: https://doi.org/10.7554/eLife.37262.016

The following figure supplement is available for figure 6:

**Figure supplement 1.** Sorting of IRSp53 I-BAR domain on self-induced membrane tubules and PIP$_2$ enrichment in these tubules.

DOI: https://doi.org/10.7554/eLife.37262.017

monomer, bringing the α-helical coiled-coils of the two monomers together (*Chambers and Bretscher, 2005*) (*Phang et al., 2016*). We show that in the presence of membranes containing PIP$_2$, both ezrinWT and ezrinTD self-assemble into densely packed, brush-like structures that tether two adjacent membranes. Upon PIP$_2$ binding, the FERM domain of ezrin interacts with the membrane; the intermolecular interaction between the N- and C-ERMAD terminals of two opposite ezrin molecules drives the formation of anti-parallel assemblies, thereby inducing the 'zipping' of two ezrin-coated membranes (*Figure 1*). Our observations agree with previous reports indicating that PIP$_2$ binding weakens the intramolecular interaction between the FERM and C-ERMAD of ezrinWT (*Pelaseyed et al., 2017*) (*Shabardina et al., 2016*). This reduction of the intramolecular interaction allows trans-intermolecular interactions between the FERM and C-ERMAD of two ezrinWT molecules on adjacent membranes, thus inducing zipping of the two adjacent membranes. The phosphomimetic mutation further reduces the intramolecular interaction (*Shabardina et al., 2016*) (*Zhu et al., 2007*), thus decreasing the intermolecular interaction, as evidenced with our dual micropipette aspiration experiments (*Figure 1G*). Our experiments demonstrate for the first time that only a low concentration of ezrin (~30 nM and ~60 nM, for ezrinTD and ezrinWT in cryoEM experiments, and down to 1% surface fraction in dual-GUV experiments) is required for the tethering activity. Moreover, we detected a local enrichment of ezrin at the GUV-GUV interface upon zipping. Thus, ezrin can zip membranes and cluster at the tethering interface at low protein density. Importantly, our experiments also provide evidence that binding to actin filaments decreases membrane tethering ability of ezrin.

## Ezrin membrane tethering capacity in cells

An estimation of the *in vivo* relative abundance of ezrin from mass spectrometry is 200 ppm (*Wang et al., 2012*) (*Geiger et al., 2012*) while that of actin is about 1% to 5% (*Berk et al., 2000*). Considering that actin concentration in cells is of the order of 500 μM (*Pollard et al., 2000*), cellular ezrin concentration is approximately 10 μM. Thus, the critical concentrations that we have measured for its tethering activity (~30 nM and ~60 nM for ezrinTD and ezrinWT, respectively) are compatible with ezrin in vivo abundance. Although the resolution of the data reported so far does not allow to discriminate whether ezrin is present in both membranes involved in fusion and whether it forms anti-parallel dimers in cells, ezrin has been reported to contribute to the fusion of vesicles in the secretory (*Yoshida et al., 2016*) (*Stanasila et al., 2006*) (*Cha et al., 2006*) (*Li et al., 2015*) and recycling pathways (*Zhou et al., 2005*) (*Tamura et al., 2005*) (*Hanzel et al., 1991*) (*Khan et al., 2013*), as well as in the fusion of phagosomes with lysosomes (*Marion et al., 2011*). Ezrin also interacts with specific receptors (*Stanasila et al., 2006*) (*Cha et al., 2006*), a sub-unit of the exocyst complex (*Li et al., 2015*) or syntaxin3 (*Yu et al., 2014*) on secretory/recycling vesicles and parietal cells' tubulo-vesicles, respectively. Moreover, ezrin is a partner of the HOPS (HOmotypic fusion and Protein Sorting) complex on endosomes, both contributing to the maturation of these organelles (*Chirivino et al., 2011*), thereby pointing to a potential role of ezrin's membrane tethering in this process. Importantly, our data show that the tethering ability of phosphorylated ezrin, at least in vitro, is modulated by its binding to actin. Thus, these findings suggest that ezrin can act as a tether in vivo, but only in local areas with a low density of actin, or when ezrin is not phosphorylated.

## The conformation of ezrin phosphomimetic mutant accounts for its positive membrane curvature sensing

We find that on PIP$_2$-containing membranes, ezrinWT and ezrinTD have distinct conformations. Consistent with previous AFM experiments on SLBs (*Shabardina et al., 2016*), the distance between the two FERM domains on tethered membranes is significantly reduced from 29 nm to 24 nm for the phosphomimetic mutant of ezrin. This difference in length of ezrinWT and ezrinTD essentially reflects a difference in the conformation of the molecules and thus possibly in their flexibility. Indeed, it was previously suggested that the α-helical region might exhibit different degrees of extensions depending on phosphorylation (*Jayasundar et al., 2012*) (*Liu et al., 2007*). The reduction of the GUV-tethering strength of ezrinTD as compared to ezrinWT indicates that a different conformation of the C-ERMAD of ezrinTD reduces its interaction with the FERM domain, and consequently the intermolecular interaction. We observe that ezrinTD, but not ezrinWT, recruits F-actin to PIP$_2$-containing membranes by forming brush-like structures at the membranes. This observation confirms that although ezrinWT is partially open upon PIP$_2$ binding (*Pelaseyed et al., 2017*) (*Shabardina et al., 2016*), its C-terminal actin-binding site remains inaccessible for F-actin binding, in agreement with previous reports (*Fievet et al., 2004*) (*Fritzsche et al., 2014*) (*Zhu et al., 2007*) (*Gautreau et al., 2000*). Finally, we observe that ezrinTD better conforms to the tubular membranes than ezrinWT and is moderately enriched onto positively curved membrane tubes *in vitro*, while ezrinWT is typically excluded. Late endosomes have a typical radius ranging between 100 to 200 nm, thus a curvature of about 2/200 nm$^{-1}$ to 2/100 nm$^{-1}$ (*Huotari and Helenius, 2011*). A clear difference was observed *in vitro* between ezrinTD and ezrinWT in this curvature range (*Figure 3E and F*). Although we cannot exclude that ezrin is recruited by a specific partner on endosomes such as the HOPS complex (*Chirivino et al., 2011*), the positive curvature sensing of the ezrin phosphomimetic mutant observed here might facilitate phosphorylated ezrin association with positively curved endosomal membranes.

## Negative membrane curvature enrichment of ezrin requires its binding to I-BAR domain proteins

Ezrin is also enriched in actin-rich membrane protrusions wherein the membranes have negative curvature. It has been proposed that PIP$_2$ binding and the specific location of the kinase LOK that phosphorylates ezrin regulate the membrane localization of ezrin (*Fievet et al., 2004*) (*Viswanatha et al., 2012*). Here we show that ezrinWT and ezrinTD do not sense negative membrane curvature and that the interaction with I-BAR domain proteins is required to facilitate the enrichment of ezrinWT or ezrinTD on membrane protrusions. We further anticipate that the direct ezrin-I-BAR interaction together with the PIP$_2$ clusters induced by the I-BAR domains (*Saarikangas et al., 2009*) (*Zhao et al., 2013*) synergistically enrich ezrin in the I-BAR domain-induced tubules. Given that IRSp53 contributes to the initiation of filopodia (*Disanza et al., 2013*), we propose that IRSp53 recruits ezrin to strengthen the binding of actin filaments to the plasma membrane that in turn facilitates filopodia growth.

In conclusion, our data corroborates the role of ezrin to maintain the mechanical cohesion of membranes with F-actin, thus facilitating actin-related cellular morphological changes. In addition, our work reveals the ability of ezrin to bridge two membranes in actin-depleted regions, or in a non-phosphorylated conformation. Furthermore, our work demonstrates the mechanisms underlying the recruitment of ezrin on different cellular membrane curvatures: the phosphomimetic mutation T567D of ezrin induces a more flexible conformation of ezrin that facilitates its interaction with positively curved membranes of some intracellular vesicles. In contrast, ezrin hijacks I-BAR domain proteins to accumulate on negatively curved membranes such as membrane protrusions and the edge of trans-endothelial cells' tunnels where ezrin is known to play a major architectural function. Interestingly, another ezrin binding partner ERM Binding Protein 50 (EBP50) was shown to interact directly with IRSp53 (*Garbett et al., 2013*) and could thus in turn recruit ezrin to microvilli. Considering that ezrin is one of the most abundant proteins at the plasma membrane (*Neisch and Fehon, 2011*), and given its actin-membrane linking function, it is presumably crucial for cells to have membrane-actin linkers that are by themselves curvature-insensitive to avoid inducing uncontrolled formation of protrusions at the plasma membrane (*Saarikangas et al., 2009*).

## Materials and methods

### Reagents

Brain total lipid extract (TBX, 131101P), brain L-α-phosphatidylinositol-4,5-bisphosphate (PIP$_2$, 840046P), 1,2-distearoyl-sn-glycero-3-phosphoethanolamine-N-[biotinyl(polyethyleneglycol)−2000] (DSPE-PEG(2000)-biotin, 880129P), 1-oleoyl-2-{6-[4-(dipyrrometheneboron difluoride)butanoyl] amino}, hexanoyl-sn-glycero-3-phosphoinositol-4,5-bisphosphate (TopFluor PIP$_2$, 810184P), L-α-phosphatidylcholine (Egg, Chicken) (EPC, 840051), and galactocerebrosides were purchased from Avanti Polar Lipids/Interchim. BODIPY-TR-C5-ceramide, (BODIPY TR ceramide, D7540), BODIPY-FL C5-ganglioside GM1 (GM1*, B13950), BODIPYFL C5-hexadecanoyl phosphatidylcholine (HPC*, D3803), Alexa Fluor 488 C5-Maleimide (AX488), Alexa Fluor 546 C5-Maleimide (AX546), Alexa Fluor 633 C5-Maleimide (AX633), were purchased from Invitrogen. GloPIPs BODIPY TMR-PtdIns(4,5)P$_2$, C16 (TMR PIP$_2$, C45M16a) was purchased from Echelon. Non-muscle actin (Actin protein,>99% pure, human platelet, APHL99) was purchased from Cytoskeleton. Streptavidin-coated polystyrene beads (SVP-30–5) were purchased from Spherotech. TRITON X-100 α-[4-(1,1,3,3-Tetramethylbutyl) phenyl]-w-hydroxy-poly(oxy-1,2-ethanediyl) (Triton-anapoe, Anapoe-X-100, anatrace) and mPEG-silane MW 2000 (mPEG-silan-2000) were purchased from Laysan Bio. Alexa Fluor 488 tagged phalloidin (AX488 phalloidin) was purchased from Interchim. β-casein from bovine milk (>98% pure, C6905) and other reagents were purchased from Sigma-Aldrich. For detecting endogenous ezrin in LLC-PK1 cells, anti-ezrin antibodies from M. Arpin laboratory was used (*Algrain et al., 1993*). For immunoprecipitation: anti-ezrin antibodies (cat# 610602, BD transduction laboratories) and anti-GFP antibodies (cat#11814460001, Roche). HRP coupled anti-mouse antibodies and Cy3 coupled anti-rabbit antibodies are from The Jackson laboratory.

### Plasmid construction

cDNA coding for either full-length human ezrin wild-type (ezrinWT) or its phosphomimetic version, ezrin T567D (ezrinTD) (*Gautreau et al., 2000*) were cloned into a pENTR vector (Invitrogen) by BxP recombination. pET28-N-His-SUMO plasmids (N-terminal His tag and SUMO fusion vector) containing the cDNA coding for either ezrinWT or ezrinTD were constructed by Gibbson Assembly (New England BioLabs) and used for the production of recombinant proteins. To ensure fluorescent maleimide labeling of recombinant ezrinWT and ezrinTD two extra cysteine residues were inserted at the C-terminal using quick-change site-directed mutagenesis (Agilent Technologies), based on the previous report (*Blin et al., 2008*). All the constructions were finally verified by sequencing.

### Protein purification and labeling

Purifications of the full-length ezrinWT, ezrinTD and the FERM domain of ezrin were performed based on a previously described procedure (*Braunger et al., 2013*). His-tagged proteins were expressed in *E. coli* Bl21 codon plus (DE3)-RIL cells induced by 0.5 mM Isopropyl β-D-1-thiogalacto-pyranoside (IPTG) for 20 hr at $20°C$. In the following, all steps were performed at $4°C$ or on ice. Bacterial pellets were collected by centrifugation at 5000 g (JLA 9.1000 rotor) for 20 min, resuspended in lysis buffer (40 mM HEPES pH 7.2, 300 mM NaCl, 5 mM β-Mercaptoethanol, one protease inhibitor tablet (complete ULTRA tablets EDTA free) and 1 mg.mL$^{-1}$ lysozyme), incubated for 30 min at 4° C and tip sonicated on ice for 4 min at 10 s/10 s, power 35%. Clear lysate was obtained by centrifugation of the lysate at 48000 g (JA 25.50 rotor) for 30 min, followed by filtered with filtration unit Stericup (0.22 μm). Purified proteins were isolated by chromatography using 2 × 1 ml Protino Ni-NTA Columns. The Ni-NTA column was equilibrated in equilibration buffer (40 mM HEPES pH 7.2, 300 mM NaCl, 5 mM β-Mercaptoethanol, 10 mM Imidazole) and bound proteins were eluted in elution buffer (40 mM HEPES pH 7.2, 300 mM NaCl, 5 mM β-Mercaptoethanol, 300 mM Imidazole) with a linear gradient from 10% to 100% of the elution buffer. To cleave the His tag, the eluted proteins were dialyzed overnight in dialysis buffer (40 mM HEPES pH 7.2, 300 mM NaCl, 5 mM β-Mercaptoethanol) containing SUMO protease. Dialyzed proteins were then purified with a Ni-NTA column equilibrated in the dialysis buffer. Finally, the eluted proteins were buffer exchanged into the labeling buffer (20 mM HEPES pH 7.2, 50 mM KCl, 0.1 mM EDTA) on a PD10 column (GE Healthcare). If not continuing the following labeling steps, pure proteins were supplemented with 2 mM β-Mercaptoethanol and 0.1% methylcellulose, snapped frozen in liquid nitrogen and stored at $−80°C$.

EzrinTD, ezrinWT and the FERM domain of ezrin labeling were performed right after the last elution step. A 5 molar excess of Alexa Fluor maleimide dyes was added to the pure proteins and allow to label at $4°C$ overnight. The labeling was quenched by supplementing 2 mM β-Mercaptoethanol in the reaction and the dye removed on a PD10 column (GE Healthcare) into the storage buffer (20 mM Tris pH 7.4, 50 mM KCl, 0.1 mM EDTA, 2 mM β-Mercaptoethanol). Finally, Alexa Fluor labeled pure proteins were supplemented with 0.1% methylcellulose, snapped frozen in liquid nitrogen and stored at $-80°C$.

Muscle actin was purified from rabbit muscle and isolated in monomeric form in G-buffer (5 mM Tris-Cl⁻, pH 7.8, 0.1 mM $CaCl_2$, 0.2 mM ATP, 1 mM DTT, 0.01% $NaN_3$) as previously described (*Spudich and Watt, 1971*). Recombinant mouse IRSp53 I-BAR domain and ABBA I-BAR domain were purified and labeled as previously described (*Prévost et al., 2015*) (*Saarikangas et al., 2009*). Gelsolin was purified as previously described (*Le Clainche and Carlier, 2004*).

## F-actin preparation

Muscle F-actin was pre-polymerized in F-buffer (5 mM Tris-Cl⁻, pH 7.8, 100 mM KCl, 0.2 mM EGTA, 1 mM $MgCl_2$, 0.2 mM ATP, 10 mM DTT, 1 mM DABCO, 0.01% $NaN_3$) for at least 1 hr at RT from Mg-ATP-actin in the presence of gelsolin at a gelsolin:actin ratio of 1:1380 to obtain F-actin with controlled lengths. As such, the resulting muscle F-actin has lengths of a few μm (*Harris and Weeds, 1984*). The same procedure was performed for non-muscle F-actin, excepted that the gelsolin:actin ratio was 1:2069. We obtained the same results as shown in *Figure 2* in the absence of gelsolin. To visualize actin, both muscle and non-muscle F-actin were labeled with equal molar amount of AX488 or AX594 phalloidin.

## Cell culture and transfection

LLC-PK1 cell line was obtained from ATCC (CCL 101; American Type Culture Collection, Rockville, MD) (*Coscoy et al., 2002*) and HeLa cell line was from B. Goud laboratory (*Echard et al., 1998*). LLC-PK1 cell line authentication was performed by M. Arpin laboratory (*Coscoy et al., 2002*) and HeLa cell was authenticated by STR. Cell lines were tested negative for mycoplasma contamination. LLC-PK1 cells and HeLa cells were cultivated at 5% $CO_2$ and at $37°C$ in DMEM supplemented with 10% (v/v) serum and 5% (v/v) penicillin/stretomycin. Transient transfection of LLC-PK1 and HeLa cells were performed with X-tremeGENE HP and X-tremeGENE 9 (Sigma-Aldrich), respectively, according to the manufacturer's instruction. Co-immunoprecipitation and mass spectrometry analysis of GFP-tagged proteins were performed by using GFP-Trap (Chromotek) according to the manufacturer's instruction.

## GUV lipid compositions and buffers

In the following experimental procedures, we use 'ezrin' to refer to ezrinTD, ezrinWT and the FERM domain of ezrin. Lipid compositions for GUVs were TBX (*Yu et al., 2006*) supplemented with 5 mole % brain $PIP_2$ , 0.025–0.5 mole% DSPE-PEG(2000)-biotin and 0.5–1 mole% BODIPY TR ceramide for ezrin tube pulling experiments. TBX supplemented with 0.1 mole% DSPE-PEG(2000)-biotin and 0.5 mole% BODIPY TR ceramide with and without 5 mole% brain $PIP_2$ for testing ezrin-$PIP_2$ binding. TBX supplemented with 5 mole% brain $PIP_2$ , 0.1 mole% DSPE-PEG(2000)-biotin and 0.5 mole% BODIPY TR ceramide for assessing ezrin-membrane binding affinity by using confocal microscopy. TBX supplemented with 5 mole% brain $PIP_2$ , and 0.5 mole% BODIPY TR ceramide for assessing ezrin-membrane binding affinity by using flow cytometry. TBX supplemented with 5 mole% brain $PIP_2$ , and 0.5 mole% BODIPY TR ceramide for GUV-ezrin tethering assay. TBX supplemented with 5 mole% brain $PIP_2$ , 0.2 mole% DSPE-PEG(2000)-biotin and 0.5 mole% BODIPY TR ceramide for biotin-streptavidin tethering assay. TBX supplemented with 5 mole% brain $PIP_2$ , and 0.5 mole% BODIPY TR ceramide for AX488 labeled ezrin and unlabeled I-BAR domain recruitment experiments. TBX supplemented with 5 mole% brain $PIP_2$ , and 0.8 mole% Rhodamine-PE ceramide for AX633 labeled ezrin and AX488 labeled I-BAR domain recruitment experiments. TBX supplemented with 4.5 mole% brain $PIP_2$ , 0.5 mole% TopFluor $PIP_2$ , 0.2 mole% DSPE-PEG(2000)-biotin and 0.5 mole% BODIPY TR ceramide for TopFluor $PIP_2$ tube pulling experiments. TBX supplemented with 4.8 mole% brain $PIP_2$ , 0.2 mole% TMR $PIP_2$ , 0.2 mole% DSPE-PEG(2000)-biotin and 0.8 mole% GM1* for TMR $PIP_2$ tube pulling experiments. TBX supplemented with 4.5 mole% brain $PIP_2$ , 0.5 mole% TopFluor $PIP_2$ ,

and 0.5 mole% BODIPY TR ceramide, and TBX supplemented with 4.2 mole% brain $PIP_2$, 0.8 mole% TopFluor $PIP_2$, and 0.5 mole% BODIPY TR ceramide for $PIP_2$-I-BAR domain induced sorting experiments. TBX supplemented with 5 mole% brain $PIP_2$, 0.8 mole% GM1*, and 0.5 mole% BODIPY TR ceramide for GM1*-I-BAR domain induced sorting experiments. DOPC supplemented with 10 mole % DGS-NTA(Ni) for preparing Ni-GUVs.

The salt buffer outside GUVs, named O-buffer, was 60 mM NaCl and 20 mM Tris pH 7.5, except for ezrin encapsulation experiments where the buffer was 300 mM NaCl and 20 mM Tris pH 7.5 to detach ezrin from binding on the outer leaflet of GUVs, and for the dilution experiments where GUVs containing ezrin were diluted in the buffer containing 60 mM NaCl, 430 mM glucose and 20 mM Tris pH 7.5. The salt buffer inside GUVs was 50 mM NaCl, 20 mM sucrose and 20 mM Tris pH 7.5, except for GUVs encapsulating ezrin where the buffer was 60 mM NaCl, 430 mM sucrose and 20 mM Tris pH 7.5, and for Ni-GUV experiments where 157 mM sucrose solution was used.

## GUV preparation

For all experiments, GUVs were prepared by electroformation on platinum electrodes under a voltage of 0.25 V and a frequency of 500 Hz overnight at 4°C in a physiologically relevant salt buffer (*Méléard et al., 2009*), except for flow cytometry experiments and for preparing Ni-GUVs. For flow cytometry experiments, GUVs was prepared by using polyvinyl alcohol (PVA) gel-assisted method in O-buffer at room temperature for 1 hr as described previously (*Weinberger et al., 2013*). Ni-GUVs were prepared by using electroformation on ITO-coated plates under a voltage of 1 V and a frequency of 10 Hz for 1 hr at room temperature in a sucrose buffer (154 mM sucrose) as described previously (*Montes et al., 2007*).

For GUVs encapsulating ezrin, 0.2–0.5 µM of ezrin was present during GUV growth. As such, the resulting GUVs have ezrin binding on both the inner and outer leaflets of the GUV membranes. Since ezrin binds to $PIP_2$ containing membranes via electrostatic interactions, screening these interactions should result in ezrin desorbing from the membranes. Indeed, when placing GUVs coated with ezrin in a high ionic strength buffer (300 mM NaCl) while keeping the osmotic pressure of the GUVs balanced, a nearly complete depletion of ezrin form GUVs was observed (*Figure 4—figure supplement 1*).

## Glass passivation

For all experiments, micropipettes for holding GUVs and microscope slides and coverslips were washed with water and ethanol followed by passivation with a β-casein solution at a concentration of 5 g.L$^{-1}$ for at least 5 min at RT. For ezrin/I-BAR domain GUV experiments, microscope slides and coverslips were cleaned by sonication with water, ethanol and acetone for 10 min, 1M KOH for 20 min, and then water for 10 min. Observation chambers were assembled and passivated with β-casein solution (5 g.L$^{-1}$ in PBS buffer) and then with mPEG-silan-2000 solution (5 mM in DMSO), each for at least 5 min.

## Measuring membrane surface fraction of proteins

We measured protein surface density on GUV membranes (number of proteins per unit area) by performing a previously established procedure (*Sorre et al., 2012*) (*Sorre et al., 2009*). We related the fluorescence intensity of the fluorescent dye used to label proteins (AX488) to that of a fluorescent lipid (BODIPY FL-C5-HPC, named HPC*). We measured fluorescence intensity of HPC* on GUV membranes at a given HPC* membrane fraction. The surface density of the protein on membranes is $n_{protein} = n_{HPC^*} / \left( \frac{I_{AX488}}{I_{HPC^*}} \right)$, where $\frac{I_{AX488}}{I_{HPC^*}}$ is the factor accounting for the fluorescence intensity difference between HPC* and AX488 at the same bulk concentration under identical image acquisition condition. The area density of HPC*, $\Phi_{HPC^*}$, can be related to its fluorescence intensity, $I_{HPC^*}^{vesicle}$, by measuring fluorescence intensities of GUVs composed of DOPC supplemented with different molar ratios of HPC* (0.04–0.16 mole%) and assuming lipid area per lipid is 0.7 nm$^2$ (1120–4480 HPC* per µm$^2$). As such, $n_{HPC^*} = A \times I_{HPC^*}^{vesicle}$, where $A$ is a constant depending on the illumination setting in the microscope. We then obtained the surface density of the protein as $n_{protein} = \left( A \times I_{protein}^{vesicle} \right) / \left( \frac{I_{AX488}}{I_{HPC^*}} \times n^* \right)$, where $n^*$ is the degree of labeling for the protein of interest. Finally, we obtained the surface fraction of the protein $\Phi_{protein} = n_{protein} \times a_{protein}$, where $a_{protein}$ is the area of a single protein on membranes.

$a_{ezrin} \cong 20$ nm$^2$ was obtained by EM analysis as shown in *Figure 1D*, and $a_{I-BAR\ domain} \cong 50$ nm$^2$ (*Millard et al., 2005*).

To obtain protein/membrane fluorescence intensity in tube pulling experiments, we manually defined a rectangular region of interest (ROI) around the membrane of a GUV or around the membrane nanotube such that the membrane/tube was horizontally located at the center of the ROI. We then obtained an intensity profile along the vertical direction of the ROI by calculating the mean fluorescence intensity of each horizontal line of the rectangle. To account for protein fluorescence inside the GUV when encapsulating or out of the GUV when injecting, the background protein intensity was obtained by calculating the average value of the mean of the first 15 intensity values from the top and the mean of the last 15 intensity values from the bottom of ROI. The membrane background intensity was obtained by calculating the mean of the first 15 intensity values from the top of ROI. Finally, the protein/membrane fluorescence intensities were obtained by subtracting the background intensity value from the maximum intensity value in the intensity profile.

## Flow cytometry experiments

To confirm with a high statistics that in the absence of PIP$_2$, ezrin does not bind to GUVs, we used flow cytometry to measure ezrin fluorescence signal as a function of ezrin bulk concentration. GUVs containing BODIPY TR ceramide were incubated with AX488 ezrin for at least 20 min at room temperature before flow cytometry measurement (BD LSRFortessaTM cell analyzer, BD Bioscience). Data was collected by BD FACSDivaTM software (BD Bioscience) and analyzed by using Flowing Software 2.5.1 (www.flowingsoftware.com). The ezrin intensity (in a.u.) on GUVs corresponds to the median value of the intensity distribution above the threshold (See *Figure 1—figure supplement 1E*)

## Cryo-electron microscopy

LUVs were prepared by detergent elimination (*Rigaud et al., 1998*). Briefly, a lipid mixture of TBX supplemented with 5 mole% PIP$_2$ was dried with argon gas and placed under vacuum for at least 3 hr. The dried lipid film was resuspended in a salt buffer (60 mM NaCl and 20 mM Tris pH 7.5) at a concentration of 1 g.L$^{-1}$. 20 mL of Triton-anapoe (10% w/v) was added into the lipid suspension and was gradually eliminated throughout the addition of biobeads in a stepwise manner to generate LUVs (20 mg of biobeads were added for an overnight incubation at $4°C$ followed by two subsequent additions of biobeads, 20 mg and 40 mg, the next morning).

LUVs were incubated with ezrin and/or F-actin at varying concentrations for at least 15 min at RT before vitrification. The samples were vitrified on copper holey lacey grids (Ted Pella) using an automated device (EMGP, Leica) by blotting the excess sample on the opposite side from the droplet of sample for 4 s in a humid environment (90% humidity). Imaging was performed on a LaB6 microscope running at 200 kV (Technai G2, FEI) and equipped with a 4K $\times$ 4K CMOS camera (F416, TVIPS). Automated data collection for 2D imaging as well as tilted series collection for cryo-tomography were carried out with the EMTools software suite.

Galactocerebroside nanotubes were prepared following the protocol given by (*Dang et al., 2005*). Briefly, galactocerebroside (Galact (β) C24 :1 Cer) was mixed with 5 mole% PIP$_2$ and 15 mole % EPC in chloroform and methanol at 10 g.L$^{-1}$. After extensive drying, the lipid film was re-suspended in 20 mM HEPES pH 7 and imidazole 200 mM at 5 g.L$^{-1}$. The solution was stirred at room temperature for one hour.

## Two dimensional image processing for EM images

To enhance the signal-to-noise ratio of the Cryo-EM images, 918 and 321 square boxes of 253 pixels were hand-picked from the ezrinTD and ezrinWT images, respectively, using the boxer tool from the EMAN software suite (*Ludtke et al., 1999*). Subsequent processing was carried out using SPIDER (*Frank et al., 1996*). After normalization of the particles, a non-biased reference-free algorithm was used to generate 10 classes. The class averages were then used individually to measure the distance between globular FERM domains within tethers. The resulting value is the average from the distances measured for each class and the error is given by the standard deviation.

To enhance the signal-to-noise ratio for the globular FERM domain, we performed two dimensional single particle analysis by selecting pieces of stacks (square boxes of 99 pixels) comprising

both bilayers and the protein material in between (900 and 320 boxes for ezrinTD and ezrinWT, respectively). Class averages were then generated by statistical analysis algorithms, 21 classes and 7 classes for ezrinTD and ezrinWT, respectively. The class averages were then used individually to measure the size of the globular FERM domains. The resulting value is the average from the sizes measured for each class and the error is given by the standard deviation.

## Cryo-electron tomography: data collection and image processing

Cryo-tomography was performed using a LaB6 microscope running at 200 kV (Technai G2, FEI) and equipped with a 4K × 4K CMOS camera (F416, TVIPS). Prior to the vitrification of the sample 10 nm gold beads were added in solution to be subsequently used as fiducials. Data collection was carried out using the EMTool (TVIPS) software suite. Tilted series were acquired from −60 to 60 degrees using a saxton angular data collection scheme. Individual images were collected with a 0.8 electrons per $Å^2$ for a total dose of less than 70 electrons per $Å^2$. Imod was primarily used for data processing and alignment for individual images. The reconstructions were performed using either IMOD (Weighted back projection) or Tomo3D (SIRT). The segmentation of the volumes was performed manually using IMOD.

## GUV-tethering experiments

For ezrin tethering, GUVs were incubated with 0.4–3 μM of AX488 ezrin in O-buffer. For ezrinTD-actin tethering experiments, GUVs were incubated with ~0.7 μM of AX488 ezrinTD for at least 5 min, followed by incubating with 0.5 μM–1.5 μM of AX594 phalloidin decorated muscle F-actin for at least 15 min. These ezrinTD-F-actin-coated GUVs were used as it is or diluted two times in O-buffer. For biotin-streptavidin tethering, GUVs incorporating 0.2 mole% biotinylated lipids were incubated with 0.4 μM or 0.8 μM of AX488 streptavidin in solution. In a typical experiment, two GUVs coated with proteins were held by micropipettes and brought into contact. We then decreased stepwise the membrane tension of the test GUV. The contact size of the two micropipette held GUVs is determined by the force balance at the contact zone: $\sigma \times \cos(\theta) = \sigma - \gamma$, where $\theta$ is the contact angle of the two GUVs, $\sigma$ is the membrane tension of the test GUV and $\gamma$ is the tethering energy at the contacting zone (see *Figure 1F* for the scheme) (*Franke et al., 2006*). The number of proteins at the tethering zone is estimated by using fluorescence signals of the proteins as described in the above section ('Measuring membrane surface fraction of proteins'), assuming all ezrin at the tethering zone contributes to the membrane tethering. This is a reasonable assumption, given that in our EM observation, we observed densely packed ezrin oligomers in-between bilayers. To obtain binding energy per ezrin bond, we assumed that ezrin tethers the membranes in its dimer form, and thus divided the tethering energy by the number of ezrin dimers at the tethering zone. The binding energy per biotin-streptavidin bond was obtained by dividing the tethering energy by the number of streptavidin at the tethering zone. Samples were observed by a X60 water immersion objective with an inverted confocal microscope (Nikon TE2000 microscope equipped with eC1 confocal system).

## Fluorescence recovery after photobleaching (FRAP) experiments on tethered GUVs

Ezrin fluorescent signals were bleached by imaging only the selected region of interest with the full laser power for ~5–10 images. After bleaching, the laser power was immediately reduced, followed by acquiring images of the two tethered GUVs. Ezrin fluorescent intensities in both the bleached region and the reference region were obtained by manually drawing a line with a width of 20 pixels perpendicularly across the membrane. We then obtained the intensity profile of the line where the x-axis of the profile is the length of the line and the y-axis is the averaged pixel intensity along the width of the line. The background intensity was obtained by calculating the mean value of the first 10 intensity values and the last 10 intensity values of the intensity profile. Finally, ezrin intensities were obtained by subtracting the background intensity from the maximum intensity value in the intensity profile. This image process was performed by using Fiji (*Schindelin et al., 2012*). To obtain FRAP curves, ezrin intensities of the bleached regions were corrected for photobleaching during image acquisition using the ezrin signals of the reference regions and then normalized by the ezrin intensity before the FRAP experiment.

## Tube pulling experiments

Tube pulling experiments were performed on a setup that comprises a Nikon C1 confocal microscope equipped with a X60 water immersion objective, micromanipulators for positioning micropipettes and optical tweezers as previously described (*Sorre et al., 2009*). To pull a tube, a GUV was held by a micropipette, brought into contact with a streptavidin-coated bead trapped by the optical tweezers, and then moved away from the bead. The tube radius $R$ was measured by using the ratio of lipid fluorescence intensity on the tube and on the GUV as $R = R_c^{TR} \times (I_{tube}/I_{vesicle})_{membrane}$, where $R_c^{TR} = 200 \pm 50$ nm is the previously obtained calibration factor for using BODIPY TR ceramide as lipid fluorescence reporter in the same setup by performing a linear fit of membrane fluorescence ratio $(I_{tube}/I_{vesicle})_{membrane}$ and lipid radii $R$ measured by $R = f/(4\pi\sigma)$, where $f$ is the force applied by the optical tweezers to sustain the tube and $\sigma$ the membrane tension controlled by the micropipette holding the GUV (*Sorre et al., 2009*) (*Prévost et al., 2015*). For experiments where GM1* lipids were used as a lipid reporter, we obtained the calibration factor $R_c^{GM1^*} = 312 \pm 15$ nm.

## Line profile along cellular protrusions

A line with a width of 4 pixels was manually drawn along a protrusion and the corresponding intensity profile was obtained by using Fiji (*Schindelin et al., 2012*).

## Immunofluorescence and immunoprecipitation

For immunofluorescence labeling, transfected LLC-PK1 cells were fixed in 3% paraformaldehyde in phosphate-buffered saline supplemented with 1 mM MgCl$_2$ and 1 mM CaCl$_2$ (PBS$^+$) for 20 min, washed with PBS$^+$ and then incubated with 0.5% Triton X100 in PBS$^+$ incubated with rabbit anti-ezrin antibodies for 1 hr, washed with PBS$^+$, incubated with secondary fluorescent antibodies (Cy3) for 1 hr, and washed with PBS$^+$.

For immunoprecipitation, HeLa cells transfected with plasmids encoding GFP, GFP-I-BAR domain or GFP-IRSp53 were washed in PBS, trypsinized, and pelleted by centrifugation at $4^\circ C$. Pelleted cells were then lysed in lysis buffer (25 mM Tris pH 7.5, 50 mM NaCl, 0.1% NP40, and protease inhibitor mix). Lysates was passed three times through a 25g syringe, incubated on ice for 1 hr to extract membrane bound proteins and then centrifuged for 10 min at 10,000 rpm to remove insoluble material and nucleus. The clean lysates were then incubated with GFP-Trap beads at $4^\circ C$ under rotation for 3 hr. The lysate-bead mixtures were washed two times in lysis buffer after centrifugation at 1850 rpm for 2 min at $4^\circ C$. The remaining buffer after the last centrifugation was removed by using a syringe. The dry lysate-bead pellet was resuspended in Laemmli buffer and processed for Western blotting following standard protocols.

## Ezrin/I-BAR domain GUV assay and fluorescence intensity quantification

For IRSp53 I-BAR domain-GUV experiments, GUVs were incubated with I-BAR domain at a concentration of 0.02–0.5 µM (containing 30 mole% of AX488 IRSp53 I-BAR domain) for at least 30 min at RT before observation. For ezrin/I-BAR domain GUV experiments, GUVs were first incubated with ezrin (0.05–2 µM) for at least 15 min at RT, and then I-BAR domain (0.05–2 µM) was added into the ezrin-GUV mixture. Samples were observed by a X100 oil immersion objective with an inverted spinning disk confocal microscope (Nikon eclipse Ti-E equipped with a EMCCD camera, QuantEM, Photometrics).

To obtain protein and membrane fluorescence intensities on a vesicle membrane or on the corresponding membrane tube to calculate $(I_{tube}/I_{vesicle})_{membrane}$ and $(I_{tube}/I_{vesicle})_{protein}$, we manually defined the ROI, a line with a width of 6 pixels drawn perpendicularly across the membrane. We then obtained the intensity profile of the line where the x-axis of the profile is the length of the line and the y-axis is the averaged pixel intensity along the width of the line. The background intensity was obtained by calculating the mean value of the first 10 intensity values and the last 10 intensity values of the intensity profile. Finally, the protein and membrane fluorescence intensities were obtained by subtracting the background intensity from the maximum intensity value in the intensity profile. This image process was performed by using Fiji (*Schindelin et al., 2012*).

To ensure we measured the protein and membrane fluorescence intensities on tubes that are in focus, we typically recorded 60 images of a GUV with 100 milliseconds exposure time and at a frame interval of about 0.5 s and manually selected tubes that were in focus.

## Statistics

All notched boxes show the median (central line), the 25th and 75th percentiles (the edges of the box), the most extreme data points the algorithm considers to be not outliers (the whiskers), and the outliers (circles).

## Acknowledgments

We thank B Goud for insightful discussions, C Le Clainche (Institute for Integrative Biology of the Cell, Gif-sur-Yvette, France) and J Pernier for providing actin and advice on actin reconstitution, F Brochard for helping on analysis of the GUV-tethering assay, F Di Federico for handling plasmids, C Prévost for her help for the optical tweezers setup and data analysis, A Di Cicco and D Levy for EM image acquisition and galcer tube preparation, N de Franceschi for helping on the FACS experiments, M Henderson for carefully reading of the manuscript. The authors greatly acknowledge the Cell and Tissue Imaging (PICT-IBiSA), Institut Curie, member of the French National Research Infrastructure France-BioImaging (ANR10-INBS-04). This work was supported by Institut Curie, Centre National de la Recherche Scientifique (CNRS), the Agence Nationale pour la Recherche (grant ANR-15-CE18-0016 to F-CT, M-CT, EL and PB), the Investments for the Future' LABEX SIGNALIFE (grant ANR-11-LABX-0028–01 for M-CT and EL), the European Research Council (EC and PB partners of the advanced grant, project 339847) and the Human Frontier Science Program Organization (RGP0005/2016 to YS, PL, F-CT, EC and PB). EC and PB groups belong to the CNRS consortium CellTiss, to the Labex CelTisPhyBio (ANR-11-LABX0038) and to Paris Sciences et Lettres (ANR-10-IDEX-0001–02). F-C Tsai was funded by the EMBO Long-Term fellowship (ALTF 1527–2014) and Marie Curie actions (H2020-MSCA-IF-2014, project membrane-ezrin-actin).

## Additional information

### Competing interests

Patricia Bassereau: Reviewing editor, *eLife*. Pekka Lappalainen: Reviewing editor, *eLife*. The other authors declare that no competing interests exist.

### Funding

| Funder | Grant reference number | Author |
| --- | --- | --- |
| European Molecular Biology Organization | ALTF 1527-2014 | Feng-Ching Tsai |
| H2020 Marie Skłodowska-Curie Actions | H2020-MSCA-IF-2014 | Feng-Ching Tsai |
| Human Frontier Science Program | RGP0005/2016 | Feng-Ching Tsai<br>Yosuke Senju<br>Pekka Lappalainen<br>Evelyne Coudrier<br>Patricia Bassereau |
| Agence Nationale de la Recherche | ANR-15-CE18-0016 | Feng-Ching Tsai<br>Meng-Chen Tsai<br>Emmanuel Lemichez<br>Patricia Bassereau |
| Investments for the Future LABEX SIGNALIFE | ANR-11-LABX-0028-01 | Meng-Chen Tsai<br>Emmanuel Lemichez |
| H2020 European Research Council | 339847 | Stephanie Miserey-Lenkei<br>Evelyne Coudrier<br>Patricia Bassereau |

The funders had no role in study design, data collection and interpretation, or the decision to submit the work for publication.

## Author contributions
Feng-Ching Tsai, Conceptualization, Formal analysis, Funding acquisition, Investigation, Visualization, Methodology, Writing—original draft, Project administration, Writing—review and editing; Aurelie Bertin, Formal analysis, Investigation, Methodology, Writing—review and editing; Hugo Bousquet, Yosuke Senju, Laura Picas, Resources, Writing—review and editing; John Manzi, Resources; Meng-Chen Tsai, Investigation; Stephanie Miserey-Lenkei, Investigation, Writing—review and editing; Pekka Lappalainen, Resources, Funding acquisition, Writing—review and editing; Emmanuel Lemichez, Funding acquisition, Writing—review and editing; Evelyne Coudrier, Patricia Bassereau, Conceptualization, Resources, Supervision, Funding acquisition, Methodology, Writing—original draft, Project administration, Writing—review and editing

## Author ORCIDs
Feng-Ching Tsai (ID) https://orcid.org/0000-0002-6869-5254
Emmanuel Lemichez (ID) http://orcid.org/0000-0001-9080-7761
Evelyne Coudrier (ID) http://orcid.org/0000-0001-6011-8922
Patricia Bassereau (ID) http://orcid.org/0000-0002-8544-6778

## Decision letter and Author response
Decision letter https://doi.org/10.7554/eLife.37262.021
Author response https://doi.org/10.7554/eLife.37262.022

## Additional files

### Supplementary files
• Transparent reporting form
DOI: https://doi.org/10.7554/eLife.37262.018

### Data availability
All data generated or analysed during this study are included in the manuscript and supporting files. Source data files have been provided for the following figures Figure 1 - figure supplement 1C, 1D, and 1G Figure 4 - figure supplement 1A and 1B.

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
