## [Decision Letter]

Thank you for submitting your article "Ezrin enrichment on curved membranes requires a specific conformation or interaction with a curvature-sensitive partner" for consideration by *eLife*. Your article has been reviewed by three peer reviewers, and the evaluation has been overseen by Anna Akhmanova as the Senior/Reviewing Editor. The reviewers opted to remain anonymous.

The reviewers have discussed the reviews with one another and the Reviewing Editor has drafted this decision to help you prepare a revised submission.

Summary:

This manuscript describes a range of biophysical and biochemical experiments where the authors set out to determine whether ezrin is a membrane curvature sensing or generating protein. Since ezrin is found in thin protrusions and binds membranes, one fundamental question is whether it contributes to the membrane deformation required to form these structures or if it is just linking the membrane curvature that is generated by other proteins to actin filaments. The authors use in vitro reconstitution to show that the wild type ezrin and a constitutively active mutant can form brushes that link adjacent membranes with different conformations. Moreover, they show how the phosphomimetic ezrin mutant, but not wild type ezrin links F-actin to membranes by exposing its FERM domain, which can act as a weak positive curvature sensor. Last, both wild type and mutant ezrin can be recruited to membrane regions with negative curvature by interacting with bona fide sensors from the IBAR domain family.

Overall this study confirms the role of ezrin as a membrane-actin linker and proves a suggested mechanism of recruitment to negatively curved membranes (interaction with I-BAR). Moreover it identifies the FERM domain as a weak positive curvature sensor.

Essential revisions:

1) The main issue with the experiments described in Figures 1 and 3 that display membrane tethering, is whether these findings help us understand how ezrin functions in cells. This could be a result of there being no actin present that would normally bind to the other side of ezrin (as shown by Figure 2). The authors need to present the findings in the bigger cellular context and provide evidence/references that support the tethering in vivo, because this is one of the main new findings.

Are the brushes observed in Figure 1 occurring at physiological concentrations? What is the minimum concentration where the authors still observe the brushes? Can the authors compare them with those reported in cells? Or could the authors discuss possible mechanisms for increased local concentrations in cells that could explain such behaviour?

2) Figure 3B top: With the images provided, it seems that the authors cannot distinguish the binding along the short axis of the tube vs. perpendicular binding, as both would look the same when imaged from the top. The authors should either reinforce that statement with tomography and averaging, or tone down their conclusions.

3) The authors should include some discussion on the significance of ezrin not being a membrane curvature protein. Many proteins are not, most are not expected to be, and it is obvious and has been thought for many years that BARs, I-BARs, etc., localise other proteins to curved areas of the cell, so a better discussion of this point would strengthen the manuscript.

4) A lot of the more detailed measurements are in supplementary figures, which makes this an awkward manuscript to read, and the authors are advised to move more of these to the main manuscript. For example, the finding that the FERM domain is a positive curvature sensor deserves moving Supplementary Figure 7D to main Figure 3.

5) Statistical analyses should be presented more clearly. Statistically significant differences should be indicated throughout the manuscript, for example, by asterisks. Where do the authors provide statistics for the measurements shown in Figure 1C? Where do the authors present error bars in Figure 1D?

---

## [Author Response]

Essential revisions:1) The main issue with the experiments described in Figures 1 and 3 that display membrane tethering, is whether these findings help us understand how ezrin functions in cells. This could be a result of there being no actin present that would normally bind to the other side of ezrin (as shown by Figure 2). The authors need to present the findings in the bigger cellular context and provide evidence/references that support the tethering in vivo, because this is one of the main new findings.

We thank the reviewers for pointing out that the presence of actin may hinder ezrin tethering ability in vivo. To test this possibility, we have performed new dual-micropipette assay in the presence of actin. We did observe a reduced efficiency of ezrin tethering, indicating that, likely in cells, the actin cortex acts as a barrier for ezrin-membrane tethering when ezrin is phosphorylated. These new results have been included in our revision (see additional data on new Figure 1G). We have also included the regulatory tethering activity of actin in the Abstract, the title of the section on F-actin (“F-actin binds to a brush-like ezrinTD assembly and reduces its membrane tethering activity”) and we added a paragraph in conclusion on this section. In addition, we would like to stress that non-phosphorylated ezrin that does not bind actin, can tether membranes independently of actin.

Moreover, there are in vivoevidences that ezrin tethering is also involved in fusion of tubulo-vesicles with plasma membrane in parietal cells. We had already mentioned this point in our previous version. A more recent study from (Yoshida et al., 2016) confirms this property. Eventually, ezrin is a partner of the HOPS complex; both contribute to endosome maturation likely by regulating endosome fusion, thereby pointing to a potential role of ezrin tethering in this process.

To emphasize the role of tethering in vivo and its regulation by actin cytoskeleton, we now have a full section on this point in the Discussion (subsection “Ezrin membrane tethering capacity in cells”).

Are the brushes observed in Figure 1 occurring at physiological concentrations? What is the minimum concentration where the authors still observe the brushes? Can the authors compare them with those reported in cells? Or could the authors discuss possible mechanisms for increased local concentrations in cells that could explain such behaviour?

Brush formation depends on 2 combined parameters: ezrin bulk concentration and the PIP_2_ concentration that sets ezrin affinity for the membrane. By titrating ezrin bulk concentrations (from 0.06 μM up to 1.2 μM) and observing ezrin tethering by Cryo-EM, we observed that membrane tethering is ezrin concentration-dependent and that ezrinTD has a greater tendency to tether membranes compared to ezrinWT (see Author response image 1). The critical concentrations for ezrin membrane tethering are approximately 30 nM and 60 nM for ezrinTD and ezrinWT, respectively. We included this result in Figure 1—figure supplement 2B (new) and in the second paragraph of the subsection “Conformations of ezrin bound to PIP2-containing membranes revealed at the nanometer scale”.

**Author response image 1. respfig1:** Percentages of tethered and non-tethered membranes deduced from cryo-EM at different ezrinTD and ezrinWT bulk concentrations. The number of samples analyzed are listed on top for each condition. n = 2 experiments per condition for protein concentration at 1.2 μM and n = 1 experiment for the other conditions.

Motivated by the referee's' question, we have searched the literature for data on the ezrin bulk concentration in eukaryotic cells. Mass spectrometry data estimates that the relative abundance of ezrin is 200 ppm (Wang et al., 2012) (Geiger et al., 2012) while that of actin is about 1 to 5% (Berk et al., 2000). Since the concentration of actin in vertebrate cells is approximately 500 μM (Pollard et al., 2000), this implies that ezrin concentration is about 10 μM. Thus, the critical concentrations that we have measured are compatible with ezrin in vivo abundance. We have added this point in the Discussion section (subsection “Ezrin membrane tethering capacity in cells”).

In addition, an obvious mechanism by which ezrin can be locally concentrated in vivo and thus could form brushes is via its interaction with PIP_2_ lipids. There are several means to induce local PIP_2_ clusters: either by inducing clusters of BAR-domain proteins (Picas et al., 2014), including IRSp53 (Saarikangas et al., 2009) (Zhao et al., 2013) or BIN1 (Picas et al., 2014), or by triggering kinases or phosphatases to produce PIP_2_. In conclusion, the brushes shown in vitro in Figure 1 could also occur in physiological conditions.

2) Figure 3B top: With the images provided, it seems that the authors cannot distinguish the binding along the short axis of the tube vs. perpendicular binding, as both would look the same when imaged from the top. The authors should either reinforce that statement with tomography and averaging, or tone down their conclusions.

To successfully perform averaging, a sample has to be remarkably organized and to display a periodic pattern. From raw images of ezrin bound to the nanotubes, one can infer that the required level of organization was not achieved. Hence averaging was left out. We also tried to perform tomography on isolated tubes. However, the low signal over noise ratio as well as the lack of organization of the proteins on the tubes preclude any clear visualization of the orientation and localization of the proteins. Considering the difficulty to acquire convincing EM images at low ezrin density, we have removed former Figure 3B and former Supplementary Figure 6. However, we think that the former Figures 3A and 3C (now Figures 3A, B and C) show a clear difference on the way that ezrin TD and WT collectively organize rigid tubes, revealing differences at the molecular level on their conformation when bound onto these membrane tubes.

3) The authors should include some discussion on the significance of ezrin not being a membrane curvature protein. Many proteins are not, most are not expected to be, and it is obvious and has been thought for many years that BARs, I-BARs, etc., localise other proteins to curved areas of the cell, so a better discussion of this point would strengthen the manuscript.

As the reviewers point out, the idea that curvature sensor proteins with BAR or I-BAR domains can recruit non-curvature sensitive partners to specific curved membrane regions has been proposed, including regulators of the actin cytoskeleton. However, by using in vitroassay, we clearly demonstrated, for the first time, that (1) ezrin does not sense negative membrane curvature, (2) the I-BAR domains are binding partners of ezrin, and (3) the I-BAR domains can recruit ezrin to tubular membranes.

Given that ezrin is abundant at the plasma membrane and its major function is to link actin to membranes, if ezrin could sense membrane curvature, it would generate a large amount of uncontrolled membrane protrusions, similarly to BAR and I-BAR domain proteins (Saarikangas et al., 2009). See our Discussion subsection “Negative membrane curvature enrichment of ezrin requires its binding to I-BAR domain proteins”, last paragraph.

Together, our study is, to our knowledge, the first one demonstrating that a protein can associate with flat, positively curved or negatively curved membranes depending on its interactions with a BAR domain protein, other interacting partners and its conformation.

We have changed the last sentence of the Introduction from "Altogether our data *reveals new mechanisms* for enriching ezrin…." into "Altogether our data *demonstrates the* mechanisms for enriching ezrin….", and similarly in the Discussion (see the aforementioned paragraph).

4) A lot of the more detailed measurements are in supplementary figures, which makes this an awkward manuscript to read, and the authors are advised to move more of these to the main manuscript. For example, the finding that the FERM domain is a positive curvature sensor deserves moving Supplementary Figure 7D to main Figure 3.

We followed the recommendations and we reduced the number of supplementary figures from 11 to 5.

Supplementary Figure 2A becomes Figure 1A, Supplementary Figure 3C is merged with Figure 1F, Supplementary Figure 4 becomes Figure 2B and 2D, Supplementary Figure 5 is removed since it had the same message as Figure 2J, Supplementary Figure 6 is completely removed (see our answer to point 2), Supplementary Figure 7B becomes Figure 3D, Supplementary Figure 7D becomes Figure 3G and Supplementary Figure 11 becomes Figure 6E.

Supplementary Figure 8A and 8C corresponding to 2 independent experiments have been merged into a single Figure 4—figure supplement 1A, the same for Supplementary Figure 8B and 8D that become Figure 4—figure supplement 1B. Supplementary Figure 2B is merged with Figure 3—figure supplement 1, and Supplementary Figure 9 and Supplementary Figure 10 are combined in Figure 6—figure supplement 1.

5) Statistical analyses should be presented more clearly. Statistically significant differences should be indicated throughout the manuscript, for example, by asterisks. Where do the authors provide statistics for the measurements shown in Figure 1C? Where do the authors present error bars in Figure 1D?

We have now included p values and corresponding asterisks on Figure 3C (former Figure 3C), Figure 1—figure supplement 1C and 1D, Figure 1—figure supplement 2A (former Figure 1E) and Figure 4—figure supplement 1A and 1B (former Supplementary Figures 8A-D). We have directly indicated the standard deviation on the scheme of Figure 1E (former Figure 1D). We also provided the statistics for the measurements shown in Figures 1C and 1D (former Figure 1B and 1C) (sizes of the globular domains and their distances between tethered membranes) both in the figure legend and in the main text (subsection “Conformations of ezrin bound to PIP2-containing membranes revealed at the nanometer scale”, second paragraph) and we refer to more details in the Materials and methods section. These distances are provided as mean ± SD both in the legends and in the main text.